# Phytochemical Analysis, Antioxidant and Enzyme-Inhibitory Activities, and Multivariate Analysis of Insect Gall Extracts of *Picea koraiensis* Nakai

**DOI:** 10.3390/molecules28166021

**Published:** 2023-08-11

**Authors:** Yanqiu Wang, Hui Sun, Xu He, Meihua Chen, Hao Zang, Xuekun Liu, Huri Piao

**Affiliations:** 1College of Pharmacy, Yanbian University, Yanji 133000, China; 2Tonghua Health School, Tonghua 134000, China; 3Green Medicinal Chemistry Laboratory, School of Pharmacy and Medicine, Tonghua Normal University, Tonghua 134002, China

**Keywords:** *Picea koraiensis* Nakai, insect gall, phytochemical analysis, biological activity, multivariate analysis

## Abstract

*Picea koraiensis* Nakai (PK) is an evergreen tree. It plays an important role in landscaping and road greening. Insect galls of PK are formed by parasitism of the adelgid *Adelges laricis*. Except for phenolics, other chemical constituents and biological activity of insect gall from PK are still unknown. Thus, here, we performed phytochemical and biological activity analyses of PK insect gall extracts, aiming to turn waste into treasure and serve human health. PK insect gall extracts were prepared using seven solvents. Antioxidant activities of the extracts were examined via antioxidant assays (radical and oxidizing substance quenching, metal chelating, and reducing power). The inhibitory activities of the extracts were determined toward the key human-disease-related enzymes *α*-glucosidase, *α*-amylase, cholinesterase, tyrosinase, urease, and xanthine oxidase. The content of numerous active constituents was high in the methanol and ethanol extracts of PK insect gall, and these extracts had the highest antioxidant and enzyme-inhibitory activities. They also showed excellent stability and low toxicity. These extracts have potential for use as stabilizers of olive and sunflower seed oils.

## 1. Introduction

Bulged parts or fruit-like structures are common on the leaves and branches of some plants. People often mistakenly think these are the fruit or seed of the plant, but they are not—these structures are called “galls”. Galls are nodules or protuberances formed by accelerating cell division and abnormal differentiation after plant tissue is stimulated by another organism. There are >2000 gall-forming organisms; gall-forming animals include insects, mites, and nematodes, while gall-forming microorganisms include bacteria, fungi, and viruses. Galls can be produced in almost all kinds of higher plants. The phenomenon of insect galls—which are galls formed after plant tissue is stimulated by feeding or oviposition by insects [1]—is very common.

Insect gall is a product of parasitic interaction between insects and plants. Insects are the beneficiary: they obtain food and a habitat from the host plant, and they can avoid predation and adverse climates. Plants are the victims of this interaction: plant growth is slowed, and plant height and leaf area are decreased. Furthermore, serious cases can cause death [2]. However, in most cases, the damage of galls to plants is limited, and the formation of plant galls is an act of self-protection to some extent. As an abnormal proliferative tissue, the content of chemical components, such as fat, protein, starch, and tannic acid, in galls is much higher than in normal plant tissues.

*Galla chinensis* are insect galls formed on the Chinese sumac tree *Rhus chinensis*; these galls are used extensively in Chinese traditional medicine because their tannin content is as high as 70–80%. Tannic acid is extracted from these galls, and its processed products are gallic acid and pyrogallic acid, which have antidiarrheal, antiperspirant, hemostatic, and wound healing effects. A decoction of *Galla chinensis* can be used to treat peptic ulcers and inflammation [3], and it is also used in the adjuvant treatment of diabetes mellitus and cancer [4].

*Picea koraiensis* Nakai (PK) is an evergreen tree in the family Pinaceae. It is distributed in the northeast Xiao Hinggan Mountains and the Jilin Mountains of China, the Russian Far East, and North Korea. It is a precious tree species that plays an important role in landscaping and road greening. Insect galls of PK endanger the shoots and buds of the tree and affect its normal growth and development, and the damaged branches dry up and die. Insect galls of PK are formed by *the* parasitism of *Adelges laricis*. *Larix kaempferi* is a secondary host in the same area [5]. The shoots and buds stimulated by the adelgid gradually grow and form spherical insect galls. The interior of the gall is composed of multiple cavities, in which there are several to dozens of *Adelges laricis*. There are “fish scale” patterns on the surface of the galls (Figure 1). The pattern is light pink at first, turns purple-red with the growth of the gall, and finally turns black. At the end of autumn, the pattern cracks, and the *Adelges laricis* drill out. On *Larix kaempferi*, *Adelges laricis* suck the sap from needles and shoots and produce large amounts of moldy branches and waxy secretions that dry; the growth of the tree is severely affected [5]. The control method is to avoid mixed or close planting of PK and to manually remove gall from PK. Removed galls are usually thrown away as waste.

In recent years, some researchers have found that phenolics are usually produced in cones as well as in galls from PK in response to insect-induced tissue damage [6,7]. Except for phenolics, other chemical constituents and biological activity of insect gall from PK are, however, unknown. Thus, here, we performed a phytochemical analysis and assessed the biological activities of insect gall extracts from PK, aiming to turn waste into treasure and serve human health.

## 2. Results and Discussion

### 2.1. Qualitative Phytochemical Analysis

Eleven classes of phytochemicals were identified in PK insect gall (Table 1). Saponins, anthraquinones, cardiac glycosides, and cyanogenic glycosides were not detected.

### 2.2. Yields

Seven solvents with distinctive polarities were selected to extract PK insect gall powder. The extraction yields ranged from 5.28 ± 0.21% to 34.32 ± 0.41% (*w*/*w*) (Table 2). The aqueous extract had the highest yield because of the abundance of water-soluble constituents (such as polyphenols, phenolics, polysaccharides, and others), followed by the methanol and ethanol extracts. Hexane extract had the lowest yield. 

### 2.3. Quantitative Phytochemical Analysis

#### 2.3.1. Total Carbohydrate Content

Carbohydrates are important constituents of animal and plant cells. They have physiological functions, such as storing and supplying energy, saving protein, and resisting the production of ketone bodies [8]. Glycosides also have various pharmacological effects, which could be used to treat neuronal diseases [9].

The total carbohydrate content (TCC) was detected in various solvent extracts of PK insect gall (Table 3). The TCC of the solvent extracts was 5.42 ± 0.31 to 325.83 ± 0.47 mg glucose equivalents (GE)/g extract. Significant differences were found among the groups: the methanol extract had the highest TCC, followed by the aqueous extract. The hexane extract had the lowest TCC. This may be because many components in the methanol extract are glycosides.

#### 2.3.2. Total Protein Content (TP_ro_C)

Plant protein has a wide range of properties and nutritional value, and it is easily digested and absorbed by the human body. It has various functions, such as immune regulation, and antioxidation and antifatigue activities [10]. Only the aqueous extract of PK insect gall contained protein, which was 313.32 ± 11.61 mg bovine serum albumin equivalents (BSAE)/g extract (Table 3).

#### 2.3.3. Total Triterpenoid Content

Triterpenoids have a variety of biological activities, such as anticancer, antiallergy, antiatherosclerosis, and antiulcer properties [11]. The total triterpenoid content (TT_ri_C) of the seven different solvent extracts of PK insect gall was determined (Table 3). Triterpenoid was not detected in the aqueous extract. The TT_ri_C of the other solvent extracts was 0.29 ± 0.03 to 1.89 ± 0.10 mg ginsenoside Re equivalents (GRE)/g extract. The highest content was in the hexane extract, followed by the dichloromethane extract. The lowest content was in the ethanol extract.

#### 2.3.4. Total Phenolic Content

Plant polyphenols are a class of active substances with good antioxidant capacity. They have antiviral and antitumor functions, and they are active in the prevention of cardiovascular disease and dementia [12]. They are widely used in the fields of cosmetics, food, and medicine [13]. The total phenolic content (TP_he_C) was determined in the seven solvent extracts of PK insect gall (Table 3). The TP_he_C ranged from 0.92 ± 0.03 to 77.11 ± 0.52 mg GAE/g extract. TP_he_C was strongly correlated with solvent polarity.

#### 2.3.5. Total Flavonoid Content

Flavonoids can prevent cell degeneration and aging, inhibit the growth of cancer cells, and control blood pressure and cholesterol; they also have good preventive effects against cardiovascular and cerebrovascular diseases [14]. The total flavonoid content (TFC) in different solvent extracts of PK insect gall was low (Table 3), ranging from 2.42 ± 0.22 to 7.52 ± 0.61 mg quercetin equivalents (QE)/g extract. The highest content was in the acetone extract, followed by the ethanol extract (there was no significant difference between the two).

#### 2.3.6. Total Phenolic Acid Content

The total phenolic acid content (TPAC) was determined in the seven solvent extracts (Table 3). The content range was 1.38 ± 0.21 to 39.92 ± 2.54 mg caffeic acid equivalents (CAE)/g extract. The content was highest in the ethanol extract, followed by the methanol extract; it was lowest in the dichloromethane extract. Similarly to TP_he_C, TPAC was strongly correlated with solvent polarity.

#### 2.3.7. Total Tannin Content, Gallotannin Content, and Condensed Tannin Content

Tannin has antibacterial and antiviral effects, and it can also be used as an antidote for alkaloid and heavy metal poisoning. It has strong reducing properties, and it can remove superoxide free radicals in the body and delay aging [15].

The gallotannin content (GC), condensed tannin content (CTC), and total tannin content (TT_an_C) of the solvent extracts of PK insect gall were determined (Table 4); these ranged from 0.36 ± 0.01 to 1.81 ± 0.22, 2.22 ± 0.23 to 38.54 ± 0.23, and from 4.32 ± 0.01 to 53.64 ± 0.22 mg CAE/g extract, respectively. The methanol extract had the highest GC and CTC, followed by the ethanol extract. The ethanol extract had the highest TT_an_C, followed by the methanol extract.

### 2.4. Antioxidant Capacity

#### 2.4.1. DPPH and ABTS

DPPH and ABTS scavenging assays are the most common methods for detecting free radical scavenging ability in vitro [16]. DPPH is a fat-soluble free radical and ABTS is a water-soluble free radical. The type of antioxidant can be judged by the difference in the scavenging activity of the antioxidant toward these two test compounds.

In experiments with DPPH, the methanol and ethanol extracts of PK insect gall showed strong antioxidant capacity (Table 4), and both were higher than butylated hydroxytoluene (BHT). In experiments with ABTS, the ethanol and methanol extracts again had high free radical scavenging ability.

#### 2.4.2. Hydroxyl Radicals, Superoxide Radicals, and Singlet Oxygen

Hydroxyl radicals, superoxide anions, and singlet oxygen are all produced in the body and result in oxidative toxicity to cells. Scavenging of these radicals reflects the capacity of antioxidants [17].

The hexane extract of PK insect gall had the highest hydroxyl radical scavenging activity, followed by the ethyl acetate and dichloromethane extracts. In superoxide anion scavenging experiments, the ethyl acetate extract had the strongest scavenging activity, while the methanol and ethanol extracts had no scavenging activity. Three extracts (water, methanol, and ethanol) had weak singlet oxygen scavenging activity (Table 4).

#### 2.4.3. FRAP and CUPRAC

The antioxidant capacity of samples can also be determined by measuring their ability to reduce iron and copper ions [18]. FRAP experiments were performed in acidic conditions, and CUPRAC experiments were performed in neutral conditions; the latter method is closer to the physiological environment, and, thus, the results can be considered more accurate.

In FRAP experiments, the methanol extract of PK insect gall had the highest antioxidant capacity, followed by the ethanol extract. They were indistinguishable from BHT in reducing ability. The results of CUPRAC experiments were consistent with those of FRAP experiments (Table 5).

#### 2.4.4. Metal Chelating

The presence of iron and copper ions accelerates the Fenton reaction and aggravates oxidative stress in vivo [19]. Therefore, the discovery of effective metal ion chelators is of great significance.

The ferrous-ion-chelating activity of the hexane extract of PK insect gall was highest, followed by the acetone and dichloromethane extracts. The copper-ion-chelating activity of the aqueous extract was the highest, followed by the ethanol and methanol extracts (Table 5).

#### 2.4.5. H_2_O_2_

H_2_O_2_ is a strong oxidant and a byproduct of human metabolism. It can cause serious damage, so direct removal of H_2_O_2_ is required for protection of the body [20]. Among the PK insect gall extracts, the ethanol extract had the greatest H_2_O_2_ scavenging activity, followed by the aqueous and acetone extracts (Table 6).

#### 2.4.6. *β*-Carotene Bleaching

*β*-Carotene is a polyene colorant that is easily oxidized, causing it to lose its yellow color [21]. In the presence of antioxidants in solution, the bleaching rate of *β*-carotene is slowed; as the capacity of an antioxidant becomes stronger, the decrease in *β*-carotene absorbance becomes slower. In experiments using the solvent extracts of PK insect gall with *β*-carotene, the acetone extract had the highest antioxidant capacity, followed by the dichloromethane and ethyl acetate extracts. There was no statistical difference among the three, and their antioxidant capacity was close to those of BHT and butyl hydroxyanisole (BHA). The aqueous extract had the weakest preventive effect on *β*-carotene bleaching (Figure 2, Table 6).

#### 2.4.7. Hypochlorous Acid

Hypochlorous acid (HClO) is an endogenous strong oxidant. It helps defend against pathogen invasion. However, excess HClO disrupts the oxidative balance of the organism, leading to the occurrence of disease [22]. Among the extracts of PK insect gall, only the methanol, ethanol, and aqueous extracts had the ability to scavenge HClO (Table 6).

#### 2.4.8. Nitric Oxide

Nitric oxide (NO) is a colorless, slightly water-soluble, fat-soluble gas. It can pass through biofilms freely in the living body. Therefore, it can participate in many biological activities and pathological and physiological processes. By acting on target molecules, NO can modify protein function and cause cellular damage. The absorbance of NO increased with time (Figure 3). The methanol extract of PK insect gall showed the best NO scavenging ability.

### 2.5. Enzyme-Inhibitory Activities

#### 2.5.1. *α*-Amylase and *α*-Glucosidase

Diabetes has become one of the major diseases that threaten human health. *α*-glucosidase and *α*-amylase promote the breakdown of food in the oral cavity and gastrointestinal tract, releasing glucose into the blood and therefore increasing blood sugar [23]. Inhibitors of these enzymes are the first choice for treatment of diabetes, but they have significant side effects. Therefore, it would be preferable to find safe inhibitors of these enzymes from natural products.

The *α*-glucosidase inhibitory activity of the methanol extract of PK insect gall was high and better than that of acarbose, which deserves further study. The *α*-amylase inhibitory activity of the acetone extract was excellent, and there was no significant difference compared with acarbose (Table 7).

#### 2.5.2. AChE and BChE

Alzheimer’s disease has a complex pathology [24]. Functional deficit of the cholinesterergic system is closely related to Alzheimer’s disease; cholinesterase inhibitors are the main direction of anti-Alzheimer’s disease drug research. The aqueous and methanol extracts of PK insect gall had AChE inhibitory activity, and the methanol extract also had BChE inhibitory activity (Table 7).

#### 2.5.3. Tyrosinase

Tyrosinase has important physiological functions. Its abnormal expression can cause melanoma and early-onset AD. Tyrosinase inhibitors have whitening functions and can be used as cosmetic additives [25]. The dichloromethane, methanol, and ethanol extracts of PK insect gall showed notable inhibitory activity toward tyrosinase (Table 7).

#### 2.5.4. Xanthine Oxidase

The main function of xanthine oxidase is to catalyze the oxidation of hypoxanthine to xanthine, which is further oxidized to uric acid. Too much uric acid in the body can lead to hyperuricemia and gout [26]. Inhibition of xanthine oxidase activity is thus a good target for the clinical treatment of hyperuricemia and gout. The acetone extract of PK insect gall had the highest xanthine oxidase inhibitory activity, followed by the methanol extract (Table 7).

#### 2.5.5. Urease

Urease, produced by *Helicobacter pylori*, catalyzes the hydrolysis of urea and increases gastric pH. This leads to the occurrence of diseases, such as gastric ulcers and atrophic gastritis. Urease inhibitors interfere with the binding of urea to the catalytic site of the enzyme, thereby decreasing its catalytic activity and reducing the harm caused by *Helicobacter pylori* infection [27]. Ethyl acetate, acetone, and hexane extracts of PK insect gall showed greater urease inhibitory activity than thiourea (Table 7), which deserves further in-depth study.

### 2.6. UHPLC–MS Analysis

The content of active ingredients in the methanol extract of PK insect gall was high, and the biological activity was good. Therefore, it was selected for further stability studies. We investigated the chemical composition of the methanol extract of PK insect gall using UHPLC–ESI–Q–TOF–MS. Twenty-five bioactive substances were identified by matching molecular ions and fragment ions with reference data (Table 8). The structures of these compounds are shown in Figure 4. The UHPLC–MS results of the methanol extract obtained in positive-ion mode are shown in Figure 5, and the MS and MS/MS results are shown in Appendix A. The UHPLC–MS results of the three reference substances (adenine, phytosphingosine, and sphinganine) obtained in positive-ion mode and the MS and MS/MS results are shown in Appendix A.

In the UHPLC–MS results, peak 1 had a [M + H]^+^ ion at *m*/*z* 337.1723, and its main fragment ion was at *m*/*z* 251.0314 ([M − C_3_H_8_N_3_]^+^) and was identified as N2-fructopyranosylarginine [28]. Peak 2 at *m*/*z* 104.1069 was tentatively assigned as choline [29]. Peak 3 had a [M + H]^+^ ion at *m*/*z* 158.1538, and its main fragment ion was at *m*/*z* 140.1438 [M − NH_3_]^+^. These results were characteristic of 4-(aminomethyl)-1-(diaminomethylene)piperidinium. Peak 4 had a [M + H]^+^ peak at *m*/*z* 136.0621 that generated a main fragment ion at *m*/*z* 118.0859 [M − NH_3_]^+^, which was characteristic of adenine [30]. The mass spectrum of peak 5 revealed an ion at *m*/*z* 140.1435. The MS^2^ spectrum of this ion showed a fragment at *m*/*z* 98.0964. The main fragment ions of peak 6 (*m*/*z* 349.1260) appeared at *m*/*z* 344.1706 [M + NH_4_]^+^, and this peak was assigned as citrusin C [31]. Peak 7 at *m*/*z* 545.1999 had MS^2^ ions at *m*/*z* 540.2458 ([M + NH_4_]^+^) and 285.1132 ([C_13_H_17_O_7_]^+^), and was identified as isolariciresinol 4′-O-beta-D-glucoside [32]. Peak 8 revealed an ion at *m*/*z* 465.1027). A fragment ion at *m*/*z* 303.0507 originated from the ion at *m*/*z* 465.1027 and was related to hexose (162 Da) loss. These results indicated that peak 8 was gossypitrin [33]. The mass spectrum of peak 9 revealed an ion at *m*/*z* 369.1537. The MS^2^ spectrum of this ion showed a fragment at *m*/*z* 167.1074. Peak 10 (*m*/*z* 449.1072) generated fragment ions at *m*/*z* 287.0551, which was related to loss of hexose. Thus, peak 10 was assigned to isoorientin [34].

Peak 11 revealed an ion at *m*/*z* 479.1180. A fragment ion at *m*/*z* 317.0665 originated from the ion at *m*/*z* 479.1180; it was related to hexose loss and it was identified as isorhamnetin-3-O-glucoside using previously reported data [35]. Peak 12 appeared at *m*/*z* 306.2071 [M + NH_4_]^+^ and it was identified as estra-1,3,5(10)-triene-3,11,17-triol based on the typical fragment ions at *m*/*z* 107.0490 ([M + H − C_11_H_18_O_2_]^+^) [36]. Peak 13 (*m*/*z* 304.1915 [M + NH_4_]^+^) had an MS^2^ ion at *m*/*z* 112.1111 ([C_6_H_8_O_2_]^+^) and was tentatively identified as 16*α*-hydroxyestrone using previously reported data [37]. The mass spectrum of peak 14 revealed an ion at *m*/*z* 466.2669. The MS^2^ spectrum of this ion showed a fragment at *m*/*z* 335.0948. Peak 15 had a [M + Na]^+^ ion at *m*/*z* 375.2142, and its main fragment ion was at *m*/*z* 309.0986 ([M − C_3_H_7_]^+^). These results were characteristic of lipoxin B4 [38]. Peak 16 had a [M + H]^+^ peak at *m*/*z* 299.1283 that generated a main fragment ion at *m*/*z* 91.0419 ([C_7_H_7_]^+^) and 77.0386 ([C_6_H_5_]^+^), which was characteristic of benzyl succinate [39]. Peak 17 at *m*/*z* 353.2299 had MS^2^ ions at *m*/*z* 301.1358 for the loss of –C_2_H_5_ and was identified as 9S,10S,11R-trihydroxy-12Z-octadecenoic acid [40]. Peak 18 at *m*/*z* 297.1856 had an MS^2^ ion at *m*/*z* 282.1617 ([M + H − CH_3_]^+^) and was identified as 5-methoxy-1,7-diphenyl-3-heptanone [41]. Peak 19 (*m*/*z* 317.2117) was tentatively identified as (10*E*,12*E*,15*E*)-9-hydroxy-10,12,15-octadecatrienoic acid based on the typical fragment ion at *m*/*z* 253.1952 ([M + H–C_3_H_6_]^+^) [42]. Peak 20 (*m*/*z* 363.2173) was identified as 11,20-dihydroxy-3-oxopregn-4-en-21-oic acid with a major MS^2^ ion at *m*/*z* 317.2122 and it was related to the loss of –COOH [43].

Peak 21 had a [M + H]^+^ peak at *m*/*z* 230.2481 that generated main fragment ions at *m*/*z* 212.2368 and 201.1640 and was related to –OH and –C_2_H_5_ loss. These results indicated that peak 21 was xestoaminol C [44]. Peak 22 had a [M + H]^+^ peak at *m*/*z* 318.3006 that generated main fragment ions at *m*/*z* 302.2220 ([M − CH_3_]^+^) and 301.2139 ([M − OH]^+^), which was characteristic of phytosphingosine [30]. And, it was consistent with the reference substance. The mass spectrum of peak 23 revealed an ion at *m*/*z* 301.2165. The MS^2^ spectrum of this ion showed a fragment at *m*/*z* 283.2068. The mass spectrum of peak 24 revealed an ion at *m*/*z* 385.2379. The MS^2^ spectrum of this ion showed a fragment at *m*/*z* 128.0620. The main fragment ion of peak 25 (*m*/*z* 287.2371) appeared at *m*/*z* 269.2265 ([M − OH]^+^), and this peak was assigned as retinol [45]. Peak 26 (*m*/*z* 337.2362) had MS^2^ ions at *m*/*z* 279.2330 ([M − C_4_H_9_]^+^), and was identified as 5,12-DiHETE. Peak 27 (*m*/*z* 387.2533) gave fragment ion at *m*/*z* 269.2280, which was correlated with the loss of –OH–C_5_H_9_O_2_+H, and was identified as 12*α*-Hydroxy-3-oxo-4,6-choladien-24-oic acid [46]. Peak 28 at *m*/*z* 302.3048 had an MS^2^ ion at *m*/*z* 303.3090 ([M + 2H]^2+^), and was tentatively assigned as sphinganine [47]. And it is consistent with reference substance. Peak 29 at *m*/*z* 317.2110 had an MS^2^ ion at *m*/*z* 318.2156 ([M + H+Na]^2+^), and was tentatively assigned as 18-hydroxy-9,11,13-octadecatrienoic acid [48]. Peak 30 (*m*/*z* 301.2160) generated fragment ions at *m*/*z* 183.1169 and 169.1012, which were related to loss of –C_7_H_11_, and –C_8_H_13_, respectively. Thus, peak 30 was assigned to *α*-linolenic acid [49].

### 2.7. Stability Studies of Methanol and Ethanol Extracts

The outcomes of stability studies of the methanol and ethanol extracts of PK insect gall are shown in Figure 6, Figure 7 and Figure 8. The TP_he_C value and ABTS scavenging activity of the extracts were broadly stable with respect to adjustments in the pH. The TP_he_C of the extracts was highest at a pH of 1 and decreased slightly with increasing pH. This is because phenolic substances can dissolve better in acidic conditions and, in strongly alkaline conditions, some phenolic components are degraded [50]. The TP_he_C of the methanol and ethanol extracts decreased with heating time; indeed, the TP_he_C of the methanol extract decreased greatly (*p* < 0.05). The ABTS scavenging activity of both extracts decreased slightly with the prolongation of heating time.

In stability experiments using an in vitro simulation of the human digestive system, the TP_he_C value of the methanol and ethanol extracts of the PK insect gall decreased gradually with time. This may be related to the effects of gastric acid, pepsin, trypsin, pancreatin, and bile on the extracts. The ABTS scavenging ability of the extracts showed the same trend as the TP_he_C.

### 2.8. Oxidative Stability Studies of Oils

Adding antioxidants improves the stability of the oils, but there are risks and hazards with using synthetic antioxidants. Therefore, the use of safe antioxidants from natural sources is desirable. Figure 9 and Figure 10 show that in K_232_ experiments, the addition of methanol extract of PK insect gall in large doses to extra virgin olive oil (EVOO) had good consequences superior to those of BHT and tertiary butylhydroquinone (TBHQ) compared with the methanol extract, with the ethanol extract having slightly lesser effects. The addition of the methanol or ethanol extracts to EVOO in K_270_ experiments showed the same trend, and their effects were similar to those of BHT. Figure 11 shows that the addition of methanol or ethanol extracts in large doses to cold-pressed sunflower oil (CPSO) was superior to the addition of BHT or TBHQ in peroxide value experiments. Figure 12 shows that in acidity values experiments, the addition of methanol extract in large doses to EVOO was effective, while the addition of methanol or ethanol extracts in small doses to CPSO was more effective.

### 2.9. Cell Viability

The cellular morphology after TM_3_ mouse Leydig cells were treated with methanol or ethanol extracts of PK insect gall for 24 and 48 h is shown in Appendix A. Overall, no cytotoxicity was observed for either the methanol or ethanol extracts after 24 h of incubation; however, the methanol extracts showed low-level toxicity after 48 h of incubation, while the ethanol extracts were comparably safe (Table 9). The methanol and ethanol extracts of PK insect gall could perhaps be used as oil stabilizers.

### 2.10. Multivariate Analysis

Through analysis of the experimental outcomes, a good correlation was found between the TCC values and the antioxidant activity (Figure 13). This implies that the carbohydrates in the methanol extract are flavonoid glycosides. Good correlations were observed between the TP_he_C, TT_an_C, TPAC, CTC, and GC values and the antioxidant activity and *α*-glucosidase and BChE inhibitory activities, as they were between the TT_ri_C value and hydroxyl radical, iron chelating, *β*-carotene, and *α*-amylase inhibitory activities (Figure 13). Good correlations were observed between the TFC value and DPPH, ABTS, and CUPRAC (Figure 13). The *α*-glucosidase and BChE inhibitory activity was linked to antioxidant capacity (Figure 14).

Univariate statistical methods were used, *p*-values < 0.05 were considered significant for all active constituents, and biological activities were assessed separately. A multiple comparative analysis applying least significant difference (LSD) post hoc analysis showed that some samples showed the same content and activity against some active constituents’ content and biological activities. Therefore, the properties of the samples depended on their active constituents. The dataset was applied to the principal component analysis (PCA). The results are depicted in Figure 15.

Three principal components (PCs), explaining 85.4% of the data variance, were identified (Figure 15D). The contributions of the active constituents and biological activities to the three PCs are expressed in Figure 15E.

PC1 summarized 59.4% of the data variance and differentiated samples in accordance with active constituents’ content and antioxidant capacities. The values for PC2 and PC3, representing other sources of variability (enzyme-inhibitory activity and TFC), were 15.4% and 10.6%, respectively. The analysis of factorial maps (Figure 15A–C) implied that the methanol and ethanol extracts were distinct from the other extracts of PK insect gall. However, the resolution between the other samples was poor.

Accordingly, the PCA outcomes were applied for hierarchical clustering analysis. The analysis of the three PCs from PCA identified four clusters (Figure 16). The first cluster, which comprised the dichloromethane and hexane extracts, displayed the highest TT_ri_C value and hydroxyl radical scavenging, iron-chelating, and tyrosinase-inhibitory activity. The second cluster, including the ethyl acetate and acetone extracts, had the highest TFC value, superoxide radical scavenging activity, and *α*-amylase, xanthine oxidase, and BChE inhibition, and it showed the greatest prevention of *β*-carotene bleaching. The third cluster, the aqueous extract, displayed the highest copper-chelating activity and AChE inhibition. The fourth cluster, composed of the methanol and ethanol extracts, exhibited high TCC, TP_he_C, TPAC, GC, CTC, and TT_an_C, high antioxidant activity, and high *α*-glucosidase and AChE inhibition.

## 3. Material and Methods

### 3.1. Reagents and Chemicals

Yeast *α*-glucosidase (EC 3.2.1.20), mushroom tyrosinase (EC 1.14.18.1), milk xanthine oxidase (EC 1.17.3.2), jack bean urease (EC 3.5.1.5), horseradish peroxidase (EC 1.11.1.7), porcine pancreatic *α*-amylase (EC 3.2.1.1), electric eel AChE (EC 3.1.1.7), horse serum BChE (EC 3.1.1.8), 3-(4,5-dimethylthiazol-2-yl)-2,5-diphenyl-2H-tetrazolium bromide (MTT), S-butyrylthiocholine chloride, and NBT were purchased from Sigma-Aldrich. Curcumin, salicylic acid, ATCI, *L*-ascorbic acid, 2,4,6-tri(2-pyridyl)-s-triazine (TPTZ), ammonium acetate (NH_4_Ac), cupric sulphate, ferrous sulfate heptahydrate (FeSO_4_·7H_2_O), copper sulphate (CuSO_4_), taurine, 4-aminoantipyrine, lipoic acid, ferulic acid, sulphanilamide, cupric chloride dihydrate (CuCl_2_·2H_2_O), phosphoric acid (H_3_PO_4_), ninhydrin hydrate, quercetin, naphthylethylenediamine dihydrochloride, D-(+)-glucose, BHT, 2,9-dimethyl-1,10-phenanthroline (Neocuproine, Nc), *α*-naphthol, iodine, TBHQ, 3,5-dinitrosalicylic acid (DNS), gelatin, potassium iodide (KI), ferric chloride (FeCl_3_), 4-nitroaniline, sodium nitrite, antimony trichloride, calcium hydroxide (Ca(OH)_2_), ABTS, copper sulfate pentahydrate (CuSO_4_·5H_2_O), phosphomolybdic acid hydrate, hydroxylamine hydrochloride, potassium hydroxide, vanillin, 3,5-dinitrobenzoic acid, phenol, dipotassium hydrogenphosphate, potassium dihydrogen phosphate, sodium dihydrogen phosphate, dibasic sodium phosphate, sodium nitroprusside dehydrate, sodium hypochlorite (NaClO) (10% active chloride), tannic acid, potassium persulfate, potassium chloride (KCl), sodium acetate, gallic acid, sodium molybdate, arbutin, L-tyrosine, urea, ginsenoside Re, phloroglucinol, potassium iodate, thiourea, hydroxyurea, xanthine, and oleanolic acid were purchased from Energy Chemical. Donepezil hydrochloride and Benedict’s Reagent were purchased from Adamas. DTNB and DPPH were purchased from Alfa Aesar. Acarbose, *β*-carotene, bromocresol green, trolox, pyrocatechol violet, sudan III, and sudan IV were purchased from TCI. p-Nitrophenyl-*α*-D-glucopyranoside, Folin and Ciocalteu’s phenol reagent (FC reagent), aluminium chloride hexahydrate (AlCl_3_·6H_2_O), linoleic acid, 3-(2-Pyridyl)-5,6-diphenyl-1,2,4-triazine-4′,4″-disulfonic acid sodium salt (Ferrozine), soluble starch, ferrous chloride tetrahydrate (FeCl_2_·4H_2_O), sodium potassium tartrate tetrahydrate (Rochelle salt), ethylenediaminetetraacetic acid disodium salt dihydrate (EDTANa_2_·2H_2_O), potassium ferricyanide (K_3_[Fe(CN)_6_]), Lead(II) acetate trihydrate, tungstosilicic acid hydrate, bismuth subnitrate, mercury(II) chloride (HgCl_2_), pepsin (32 U/mg), pancreatin, bovine bile extract, magnesium acetate, sodium thiosulfate standard solution (0.1 M), potassium hydroxide standard solution (0.1 M), phenolphthalein, tween 40, and 1,3-dinitrobenzene were purchased from Xiya Reagent. Concentrated sulfuric acid (H_2_SO_4_), phenol, sodium carbonate (Na_2_CO_3_), methanol, ethanol, acetone, ethyl acetate, dichloromethane, hexane, dimethyl sulfoxide (DMSO), petroleum ether (60–90 °C), sodium hydroxide (NaOH), concentrated hydrochloric acid (HCl), sodium chloride (NaCl), magnesium powder, acetic acid, ammonium hydroxide (NH_3_·H_2_O), acetic anhydride, 30% hydrogen peroxide (H_2_O_2_), formaldehyde, and 3% bromine water were purchased from Sinopharm. All reagents and solvents used were analytical grade. Litmus paper blue was purchased from Tianjin Jinda Chemical Reagent Co., Ltd., Tianjin, China. BCA kit was purchased from Beyotime. High-glucose Dulbecco’s modified Eagle’s medium (HG-DMEM) and 100× penicillin–streptomycin solution were purchased from Hyclone. Fetal bovine serum (FBS) was purchased from Bioind. TM_3_ mouse leydig cells were purchased from the Cell Bank of the Chinese Academy of Sciences. Trypsin (2500 U/mg) was purchased from Aladdin. Extra virgin olive oil (EVOO) was purchased from MUELOLIVA. Cold-pressed sunflower oil (CPSO) was purchased from Daodaoquan Grain and Oil Co., Ltd., Yueyang, China. Adenine (reference substance) was purchased from Shanghai yuanye Bio-Technology Co., Ltd., Shanghai, China. Sphinganine (reference substance) was purchased from Beijing Solarbio Science & Technology Co., Ltd., Beijing, China. Phytosphingosine (reference substance) was purchased from Chengdu Alfa Biotechnology Co., Ltd., Chengdu, China.

### 3.2. Materials

Insect gall of PK was gathered (voucher specimen number: 2021-07-04-001) in Tonghua (latitude N 41°44′43.15″, longitude E 125°59′13.52″, altitude 439.8 m, Jilin Province, China) in July 2021. A plant taxonomist (Professor Junlin Yu) confirmed the identification of the specimen. The voucher specimen is stored in the Herbarium of Tonghua Normal University.

### 3.3. Preparation of Different Extracts of PK Insect Gall for Quantitative Phytochemical Analysis and UHPLC–MS Analysis

The collected samples of PK insect gall were dried in a cool ventilated place and pulverized to powder. The powder (20 g) was added to a single-neck round-bottomed flask (glass, 500 mL), followed by the addition of 200 mL of various solvents (water, methanol, ethanol, acetone, ethyl acetate, dichloromethane, or hexane) and refluxing using a hotplate magnetic stirrer employing methyl silicone oil as the heating medium for 6 h at the respective boiling points of the solvents. The extracts were filtered through a Whatman No.1 filter paper and evaporated under reduced pressure at <50 °C until dry using a rotary evaporator. All dried extracts were weighed and stored at −20 °C until use. The yield was calculated as % yield = (weight of dry extract/initial weight of dry sample) × 100.

### 3.4. Qualitative Phytochemical Analysis

Qualitative phytochemical analysis of 15 types of chemical components was performed in accordance with our previous methods [51].

### 3.5. Quantitative Phytochemical Analysis

The TCC, TP_ro_C, TT_ri_C, TP_he_C, TFC, TPAC, TT_an_C, CTC, and GC were determined in accordance with our previous methods [51,52]; the specific experimental steps are as follows.

#### 3.5.1. Determination of TCC

Briefly, 250 μL of extract of PK insect gall in distilled water, 125 μL of phenol solution (5%), and 625 μL of H_2_SO_4_ were mixed in an Eppendorf tube and incubated for 30 min. Subsequently, 200 μL of the sample was pipetted from each Eppendorf tube onto a microplate. A calibration curve was produced based on glucose (0–200 mg/L) as a standard. The absorbance of the sample was recorded at 490 nm against a blank sample consisting of extract of PK insect gall with distilled water. The mean of three readings was used, and TCC was expressed in milligrams of GE/g of extract of PK insect gall.

#### 3.5.2. Determination of TP_ro_C

Briefly, 200 μL of bicinchoninic acid working solution and 20 μL of extract of PK insect gall in distilled water were mixed in a microplate and incubated at 37 °C for 30 min. A calibration curve was produced based on bovine serum albumin (0–500 mg/L) as a standard. The absorbance of the sample was recorded at 562 nm against a blank sample consisting of extract of PK insect gall with distilled water. The mean of three readings was used, and TP_ro_C was expressed in milligrams of BSAE/g of extract of PK insect gall.

#### 3.5.3. Determination of TT_ri_C

Briefly, 180 μL of extract of PK insect gall in acetic anhydride and 20 μL of H_2_SO_4_ were mixed in a microplate and incubated at room temperature for 10 min. A calibration curve was produced based on ginsenoside Re (0–400 mg/L) as a standard. The absorbance of the sample was recorded at 350 nm against a blank sample consisting of extract of PK insect gall with acetic anhydride. The mean of three readings was used, and TT_ri_C was expressed in milligrams of GRE/g of extract of PK insect gall.

#### 3.5.4. Determination of TP_he_C

Briefly, 100 μL of Folin and Ciocalteu’s phenol reagent (FC reagent) (1 M) and 200 μL of extract of PK insect gall in distilled water were mixed in an Eppendorf tube and incubated for 5 min. Subsequently, 500 μL of Na_2_CO_3_ solution (20%) was added and allowed to stand at room temperature for 40 min in the dark (with mixing every 10 min). Subsequently, 200 μL of the sample was pipetted from each Eppendorf tube onto a microplate. A calibration curve was produced based on gallic acid (0–100 mg/L) as a standard. The absorbance of the sample was recorded at 750 nm against a blank sample consisting of extract of PK insect gall with distilled water and Na_2_CO_3_. The mean of three readings was used, and TP_he_C was expressed in milligrams of GAE/g of extract of PK insect gall.

#### 3.5.5. Determination of TFC

Briefly, 100 μL of AlCl_3_ (2%) in methanol and 100 μL of extract of PK insect gall in methanol were mixed in a microplate and incubated at room temperature for 10 min. A calibration curve was produced based on quercetin (0–100 mg/L) as a standard. The absorbance of the sample was recorded at 415 nm against a blank sample consisting of extract of PK insect gall with methanol. The mean of three readings was used, and TFC was expressed in milligrams of QE/g of extract of PK insect gall.

#### 3.5.6. Determination of TPAC

Briefly, 20 μL of extract of PK insect gall in distilled water, 20 µL of Arnow reagent, 20 µL of HCl solution (0.1 M), 120 µL of distilled water, and 20 µL of NaOH solution (1 M) were mixed in a microplate and recorded immediately at 490 nm against a blank sample (Arnow reagent was replaced with distilled water). A calibration curve was produced based on caffeic acid (0–100 mg/L) as a standard. The mean of three readings was used, and TPAC was expressed in milligrams of CAE/g of extract of PK insect gall.

#### 3.5.7. Determination of TT_an_C

Briefly, 200 μL of FC reagent (1 M) and 200 μL of extract of PK insect gall in distilled water were mixed in an Eppendorf tube and incubated for 5 min. Subsequently, 100 μL of Na_2_CO_3_ solution (20%) and 1500 μL of distilled water were added and allowed to stand at room temperature for 30 min in the dark (with mixing every 10 min). Subsequently, 200 μL of the sample was pipetted from each Eppendorf tube onto a microplate. A calibration curve was produced based on tannic acid (0–200 mg/L) as a standard. The absorbance of the sample was recorded at 725 nm against a blank sample consisting of extract of PK insect gall with distilled water and Na_2_CO_3_. The mean of three readings was used, and TT_an_C was expressed in milligrams of TAE/g of extract of PK insect gall.

#### 3.5.8. Determination of GC

Briefly, 875 µL of extract of PK insect gall in methanol and 375 µL of saturated KIO_3_ solution were mixed in an Eppendorf tube and incubated at 15 °C for 120 min. A calibration curve was produced based on gallic acid (0–400 mg/L) as a standard. The absorbance of the sample was recorded at 550 nm against a blank sample (KIO_3_ was replaced with distilled water). The mean of three readings was used, and GC was expressed in milligrams of GAE/g of extract of PK insect gall.

#### 3.5.9. Determination of CTC

Briefly, 4 mg of phloroglucinol was added to 2 mL of extract of PK insect gall in distilled water. Subsequently, 1 mL of HCl solution and 1 mL of formaldehyde solution were added and mixed in an Eppendorf tube and incubated at room temperature overnight. The precipitate was separated by filtration, and the unprecipitated phenolics were measured in the filtrate according to the method of TP_he_C.

### 3.6. Antioxidant Activity Assays

All antioxidant activity assays (including those using DPPH, ABTS, hydroxyl radicals, superoxide radicals, FRAP, CUPRAC, metal chelating, H_2_O_2_, *β*-carotene bleaching, and NO) were performed in accordance with our previous methods [51,52]; the specific experimental steps are as follows.

#### 3.6.1. DPPH Assay

Briefly, 100 µL of extract of PK insect gall in methanol and 100 µL of DPPH in methanol (50 µM) were mixed in a microplate and allowed to stand at room temperature for 20 min in the dark. The absorbance of the sample was recorded at 515 nm. Trolox was used as a positive reference, and its standard curve is y = 11.595x + 0.1821. The half-maximal inhibitory concentration (IC_50_) values were calculated and expressed as the mean ± standard deviation (SD) in μg/mL.

#### 3.6.2. ABTS Assay

Briefly, 190 μL of diluted ABTS solution and 10 μL of extract of PK insect gall in DMSO were mixed in a microplate and incubated for 20 min in the dark. The absorbance of the sample was recorded at 734 nm. Trolox was used as a positive reference, and its standard curve is y = 0.8026x + 11.878. The IC_50_ values were calculated and expressed as the mean ± SD in μg/mL.

#### 3.6.3. Hydroxyl Radical Assay

Briefly, 50 µL of extract of PK insect gall in DMSO, 50 µL of FeSO_4_ solution (3 mM), and 50 µL of H_2_O_2_ solution (3 mM) were mixed in a microplate and incubated for 10 min. After, 50 µL of salicylic acid solution (6 mM) was added and incubated at room temperature for 30 min in the dark. The absorbance of the sample was recorded at 492 nm. Trolox was used as a positive reference, and its standard curve is y = 0.0327x + 35.047. The IC_50_ values were calculated and expressed as the mean ± SD in μg/mL.

#### 3.6.4. Superoxide Radical Assay

Briefly, 45 µL of extract of PK insect gall in DMSO (10 mg/mL), 15 µL of *p*-nitroblue tetrazolium chloride (NBT) in DMSO (1 mg/mL), and 150 µL of NaOH in DMSO (50 μM) were mixed in a microplate, and the absorbance of the sample was recorded immediately at 560 nm against a blank sample (NBT was replaced with DMSO). Curcumin was used as a positive reference, and its standard curve is y = 84.38239x/122.02931 + x. The scavenging activity was expressed as % scavenging rate and was calculated as follows:$$\%scavenging=\left(1-\frac{\Delta A_{sample}}{\Delta A_{control}}\right)\times 100\%.$$

#### 3.6.5. FRAP Assay

Briefly, 20 µL of extract of PK insect gall in DMSO and 180 µL of FRAP reagent were mixed in a microplate and incubated at 37 °C for 30 min in the dark. A calibration curve was produced based on FeSO_4_ (0–600 mg/L) as a standard. The absorbance of the sample was recorded at 595 nm. The standard curve of ferrous ion is y = 4.416x + 0.087. Trolox was used as a positive reference. The FRAP was expressed as the Trolox Equivalent Antioxidant Capacity (TEAC_FRAP_).

#### 3.6.6. CUPRAC Assay

Briefly, 20 µL of CuCl_2_ solution (100 mM), 50 µL of neocuproine in 96% ethanol (7.5 mM), 50 µL of NH_4_Ac solution, 20 µL of extract of PK insect gall in DMSO, and 30 µL of distilled water were mixed in a microplate and incubated at 50 °C for 20 min. This mixture was allowed to stand at room temperature for 10 min. The absorbance of the sample was recorded at 450 nm. Trolox was used as a positive reference, and its standard curve is y = 2.8264x + 0.0462. The CUPRAC was expressed as the Trolox Equivalent Antioxidant Capacity (TEAC_CUPRAC_).

#### 3.6.7. Iron Chelating Assay

Briefly, 50 µL of extract of PK insect gall in methanol, 110 µL of ultra-pure water, and 20 µL of FeCl_2_ solution (0.5 mM) were mixed in a microplate and incubated for 5 min. Subsequently, 20 µL of ferrozine solution (2.5 mM) was added and incubated for 10 min. The absorbance was recorded at 562 nm against a blank sample (ferrozine solution was replaced with water). Ethylenediaminetetraacetic acid disodium salt (EDTANa_2_) was used as a positive reference, and its standard curve is y = 2142.2x + 26.6. The IC_50_ values were calculated and expressed as the mean ± SD in μg/mL.

#### 3.6.8. Copper Chelating Assay

Briefly, 40 µL of extract of PK insect gall in ultra-pure water, 140 µL of acetic acid-sodium acetate buffer solution (pH 6.0, 50 mM), and 10 µL of CuSO_4_ solution (5 mM) were mixed in a microplate and incubated for 30 min. Subsequently, 10 µL of pyrocatechol violet solution (4 mM) was added and incubated for 30 min. The absorbance was recorded at 632 nm against a blank sample (pyrocatechol violet was replaced with water). EDTANa_2_ was used as a positive reference, and its standard curve is y = 214.5x + 6.9643. The IC_50_ values were calculated and expressed as the mean ± SD in μg/mL.

#### 3.6.9. H_2_O_2_ Assay

Briefly, 70 µL of phenol solution (pH 7.0, 12 mM, in 84 mM phosphate buffer (PBS)), 20 µL of 4-aminoantipyrine solution (pH 7.0, 0.5 mM, in 84 mM PBS), 32 μL of H_2_O_2_ solution (pH 7.0, 0.7 mM, in 84 mM PBS), 8 µL of horseradish peroxidase (EC 1.11.1.7) solution (pH 7.0, 1 U/mL, in 84 mM PBS), and 70 µL of extract of PK insect gall (pH 7.0, in 84 mM PBS) were mixed in a microplate, and the absorbance of the sample was recorded immediately at 504 nm against a blank sample (phenol solution was replaced with PBS). Gallic acid was used as a positive reference, and its standard curve is y = 0.3898x + 25.193. The IC_50_ values were calculated and expressed as the mean ± SD in μg/mL.

#### 3.6.10. Singlet Oxygen Assay

Briefly, 40 µL of extract of PK insect gall (pH 7.4, in 45 mM PBS), 50 µL of N,N-dimethyl-4-nitrosoaniline (pH 7.4, 0.2 mM, in 45 mM PBS), 20 μL of histidine solution (pH 7.4, 0.1 mM, in 45 mM PBS), 20 µL of NaClO solution (pH 7.4, 0.1 mM, in 45 mM PBS), 20 μL of H_2_O_2_ (pH 7.4, 0.1 mM, in 45 mM PBS), and 50 µL of PBS (pH 7.4, 45 mM) were mixed in a microplate and allowed to stand at room temperature for 40 min. The absorbance of the sample was recorded at 440 nm against a blank sample (extract of PK insect gall was replaced with PBS). Ferulic acid was used as a positive reference, and its standard curve is y = 41.354 − 42.868. The IC_50_ values were calculated and expressed as the mean ± SD in μg/mL.

#### 3.6.11. HClO Assay

HClO was freshly prepared by adjusting the pH of 1% (*v*/*v*) of NaClO to 6.2 with 1% H_2_SO_4_. The concentration of HClO was determined by reading the absorbance at 235 nm and using the molar extinction coefficient of 100 M^−1^ cm^−1^. Briefly, 20 microliters of AR extract aqueous solution, 20 µL of 150 mM taurine aqueous solution, 20 µL of 0.5 mM HClO solution, and 140 μL of PBS (pH 7.4, 50 mM) were mixed in a microplate and incubated for 10 min. Subsequently, 2 µL of 2 M KI aqueous solution was added and mixed. The absorbance was recorded at 350 nm against a blank sample (taurine and HClO were replaced with water). Trolox was used as a positive reference, and its standard curve is y = 0.3007x + 11.474. IC_50_ values were calculated and expressed as the mean ± SD in μg/mL.

#### 3.6.12. *β*-Carotene Bleaching Assay

Briefly, *β*-carotene solution was prepared by dissolving β-carotene (2 mg) in CHCl_3_ (10 mL). Then, 2 mL of the solution was pipetted into a flask and vortex mixed with linoleic acid (40 mg) and Tween 40 (400 mg). After the removal of CHCl_3_, 100 mL of oxygenated ultra-pure water was added, and the emulsion was shaken vigorously. Aliquots (2.4 mL) of the emulsion were pipetted into different test tubes containing 0.1 mL of extract of PK insect gall in methanol (5 mg/mL). BHT and BHA were used as positive controls. In the control group, the extract of PK insect gall was replaced with water. When the sample was added to the emulsion, it was recorded as t = 0 min. The tubes were capped and placed in a water bath at 60 °C. The absorbance was recorded at 470 nm every 15 min until 120 min. The antioxidant activity coefficient (*AAC*) was calculated according to the following equation:$$AAC=\frac{A_{A(120)}-A_{C(120)}}{A_{C(0)}-A_{C(120)}}\times 1000$$
where *A_A_*_(120)_ is the absorbance of the antioxidant at 120 min, *A_C_*_(120)_ is the absorbance of the control at 120 min, and *A_C_*_(0)_ is the absorbance of the control at 0 min.

#### 3.6.13. NO Assay

Briefly, 3 mL of extract of PK insect gall in methanol (1 mg/mL) and 3 mL of sodium nitroprusside solution (pH 7.4, 5 mM, in 0.1 M PBS) were mixed in an Eppendorf tube and incubated at 25 °C for 150 min. At intervals, 100 μL of the sample was pipetted from each Eppendorf tube onto a microplate containing 100 µL of Griess reagent. In the control group, the extract of PK insect gall was replaced with methanol. The absorbance was recorded at 546 nm against a blank sample (the Griess reagent was replaced with distilled water). Curcumin (0.1 mg/mL) was used as a positive reference.

### 3.7. Enzyme Inhibition Assays

All enzyme inhibition assays (including *α*-glucosidase, *α*-amylase, tyrosinase, urease, AChE, BChE, and xanthine oxidase) were performed in accordance with our previous methods [50]; the specific experimental steps are as follows.

#### 3.7.1. *α*-Glucosidase Inhibition Assay

Briefly, 20 μL of extract of PK insect gall in DMSO (10 mg/mL) and 100 µL of yeast *α*-glucosidase (EC 3.2.1.20) solution (pH 6.9, 0.1 U/mL, in 0.1 M PBS) were mixed in a microplate and incubated at 25 °C for 10 min. Subsequently, 50 μL of p-nitrophenyl-*α*-D-glucopyranoside solution (pH 6.9, 5 mM, in 0.1 M PBS) was added and incubated at 25 °C for 5 min. Before and after incubation, the absorbance was recorded at 405 nm. In the control group, the extract of PK insect gall was replaced with DMSO. Acarbose was used as a positive reference. The inhibitory activity was expressed as % inhibition and was calculated as follows:$$\%inhibition=\left(1-\frac{\Delta A_{sample}}{\Delta A_{control}}\right)\times 100\%$$

#### 3.7.2. *α*-Amylase Inhibition Assay

Briefly, 20 μL of extract of PK insect gall in DMSO, 80 µL of distilled water, and 100 µL of porcine pancreatic *α*-amylase (EC 3.2.1.1) solution (pH 6.9, 4 U/mL, in 20 mM PBS) were mixed in an Eppendorf tube and incubated at 25 °C for 5 min. Subsequently, 200 µL of 0.5% starch solution (pH 6.9, in 20 mM PBS) was added and incubated at 25 °C for 3 min. Then, 200 µL of the mixture was removed from the Eppendorf tube and added to a separate Eppendorf tube containing 100 µL of 3,5-dinitrosalicylic acid reagent solution and placed in a 85 °C water bath. After 15 min, the mixture was removed from the water bath and diluted with 900 µL of distilled water. The absorbance was recorded at 540 nm. In the control group, the extract of PK insect gall was replaced with DMSO. In the blank group, the enzyme solution was replaced with PBS. Acarbose was used as a positive reference. The inhibitory activity was expressed as % inhibition and was calculated as follows:$$\%inhibition=\left(1-\frac{A_{sample}-A_{blank}}{A_{control}}\right)\times 100\%$$

#### 3.7.3. AChE Inhibition Assay

Briefly, 20 μL of extract of PK insect gall in 10% DMSO (1 mg/mL), 120 µL of PBS (pH 8.0, 0.1 M), and 20 µL of AChE (EC 3.1.1.7) solution (pH 8.0, 0.8 U/mL, in 0.1 M PBS) were mixed in a microplate and incubated at 25 °C for 15 min. Subsequently, 20 μL of acetylthiocholine iodide (ATCI) solution (pH 8.0, 1.78 mM, in 0.1 M PBS) and 20 μL 5,5′-dithiobis-(2-nitrobenzoic acid) (DTNB) solution (pH 8.0, 1.25 mM, in 0.1 M PBS) were added and incubated at 25 °C for 5 min. Before and after incubation, the absorbance was recorded at 405 nm. In the control group, the extract of PK insect gall was replaced with 10% DMSO. Donepezil was used as a positive reference. The inhibitory activity was expressed as % inhibition and was calculated as same as Section 3.7.1. 

#### 3.7.4. BChE Inhibition Assay

Briefly, 20 μL of extract of PK insect gall in 10% DMSO (1 mg/mL), 120 µL of PBS (pH 8.0, 0.1 M), and 20 µL of BChE (EC 3.1.1.8) solution (pH 8.0, 0.8 U/mL, in 0.1 M PBS) were mixed in a microplate and incubated at 25 °C for 15 min. Subsequently, 20 μL of S-butyrylthiocholine chloride solution (pH 8.0, 0.4 mM, in 0.1 M PBS) and 20 μL of DTNB solution (pH 8.0, 1.25 mM, in 0.1 M PBS) were added and incubated at 25 °C for 5 min. Before and after incubation, the absorbance was recorded at 405 nm. In the control group, the extract of PK insect gall was replaced with 10% DMSO. Donepezil was used as a positive reference. The inhibitory activity was expressed as % inhibition and was calculated as same as Section 3.7.1.

#### 3.7.5. Tyrosinase Inhibition Assay

Briefly, 100 μL of L-tyrosine solution (pH 6.8, 5 mM, in 0.1 M PBS), 20 µL of PBS (pH 6.8, 0.1 M), and 40 µL of extract of PK insect gall in 10% DMSO (5 mg/mL) were mixed in a microplate. Subsequently, 40 μL of mushroom tyrosinase (EC 1.14.18.1) solution (pH 6.8, 200 U/mL, in 0.1 M PBS) was added and incubated at 37 °C for 20 min. The absorbance was recorded at 450 nm. In the control group, the extract of PK insect gall was replaced with 10% DMSO. In the blank group, the enzyme solution was replaced with PBS. Arbutin was used as a positive reference. The inhibitory activity was expressed as % inhibition and was calculated was calculated as same as Section 3.7.2. 

#### 3.7.6. Urease Inhibition Assay

Briefly, 60 μL of urea solution (pH 7.4, 100 mM, in 0.01 M PBS), 15 µL of jack bean urease (EC 3.5.1.5) solution (pH 7.4, 5 U/mL, in 0.01 M PBS), and 15 µL of extract of PK insect gall in PBS (pH 7.4, 0.01 M) were mixed in a microplate and incubated at 37 °C for 30 min. Subsequently, 60 μL of phenol reagents and 60 μL of alkali reagent were added and incubated at 37 °C for 30 min. The absorbance was recorded at 625 nm. In the control group, the extract of PK insect gall was replaced with PBS. In the blank group, the enzyme solution was replaced with PBS. Thiourea was used as a positive reference. The inhibitory activity was expressed as % inhibition and was calculated as same as Section 3.7.2. 

#### 3.7.7. XO Inhibition Assay

Briefly, 50 μL of extract of PK insect gall in PBS (pH 7.5, 5 mg/mL, 0.07 M), 35 µL of PBS (pH 7.5, 0.07 M), and 30 µL of XO (EC 1.17.3.2) solution (pH 7.5, 0.01 U/mL, in 0.07 M PBS) were mixed in a quartz microplate and incubated at 25 °C for 15 min. Subsequently, 60 μL of xanthine solution (pH 7.5, 150 μM, in 0.07 M PBS) was added, and the solution was incubated at 25 °C for 30 min. The reaction was stopped by adding 25 µL of 1 M HCl. The absorbance was recorded at 290 nm. In the control group, the extract of PK insect gall was replaced with PBS. The enzyme solution was added to the microplate after adding HCl. In the blank group, the enzyme solution was replaced with PBS. Allopurinol was used as a positive reference. The inhibitory activity was expressed as % inhibition and was calculated as same as Section 3.7.2. 

### 3.8. UHPLC–MS

Methanol extract of PK insect gall was analyzed using UHPLC (Agilent 1290 system, Santa Clara, CA, USA) with Q-TOF MS (Agilent 6545 system, Santa Clara, CA, USA). A ZORBAX SB-C_18_ column (150 × 3.0 mm, 1.8 µm; Agilent) was used. The column temperature was set to 40 °C. The mobile phase was a mixture of 0.1% formic acid in water (solvent A) and a mixture of 0.1% formic acid in acetonitrile (solvent B) at a flow-rate of 0.4 mL/min. Linear gradient elution was applied (0–1 min, 95% A; 1–30 min, 95–70% A; 30–50 min, 70–30% A; 50–56 min, 30–1% A; 56–60 min, 1% A). The extract was diluted to 1 mg/mL with methanol and filtered using a 0.22-µm membrane before use. The sample injection volume was 5 µL. The Q-TOF-MS (Agilent) was operated in positive-ion mode with scan range *m*/*z* 100–1700. Data were recorded and analyzed with Qualitative Analysis software (version B.07.00, Agilent, Santa Clara, CA, USA).

### 3.9. Stability of Methanol and Ethanol Extracts

The pH stability, thermal stability, and stability in a gastrointestinal tract model system of the extracts were determined in accordance with our previous methods [50]; the specific experimental steps are as follows.

#### 3.9.1. pH Stability

The stability in acidic and basic environments was investigated using a methanol extract or ethanol extract of PK insect gall dissolved in deionized water with the pH adjusted to 1, 3, 5, 7, 9, or 11 using 1 M HCl or 1 M NaOH. The final concentration of methanol extract or ethanol extract was 50 mg/mL. After incubation at room temperature for 1 h, the pH of the mixture was adjusted to 7, and the TP_he_C and the ABTS scavenging abilities were examined.

#### 3.9.2. Thermal Stability

To evaluate the thermal stability, methanol extract or ethanol extract of PK insect gall dissolved in deionized water (50 mg/mL, pH 7) was placed in test tubes with screw caps. The test tubes were placed in a boiling water bath (100 °C). Samples were removed after 0, 15, 30, 60, 120, 180, and 240 min and cooled in an ice water bath. The TP_he_C and the ABTS scavenging abilities were examined.

#### 3.9.3. Modeling of the Stability in the Gastrointestinal Tract

First, 100 mL of methanol extract or ethanol extract of PK insect gall in distilled water (5 mg/mL) were mixed with 10 mL of PBS (pH 6.8, 10 mM) and incubated at 37 °C for 2 min (oral condition). Then, 0.5 mL of 1 M HCl-KCl buffer (pH 1.5) and 5 mL of pepsin solution (pH 1.5, 32 U/mL in 1 M HCl-KCl buffer) were added to the samples. The mixtures were incubated at 37 °C for 60 min (stomach condition). Thereafter, into the mixture was added 1 mL of 1 M NaHCO_3_ together with 1 mL of mixture of bile and pancreatic juice (pH 8.2, 10 mg/mL of pancreatin, 14,600 U/mL of trypsin, 13.5 mg/mL of bile extract in 10 mM PBS), and the pH was adjusted to 6.8. The mixtures were incubated at 37 °C for 3 h (duodenal condition). The results were used for the determination of TP_he_C, and the ABTS scavenging abilities of methanol extract or ethanol extract during simulated gastrointestinal digestion were taken at 0, 0.5, and 1–4 h.

### 3.10. Oxidative Stability of Oils

The oxidative stability of EVOO and CPSO was evaluated according to our previous methods [50]; the specific experimental steps are as follows.

EVOO and CPSO were placed in separate flasks. Methanol extract or ethanol extract of PK insect gall was added to the EVOO and CPSO flasks at concentrations of 1000 and 250 μg/g. To compare with the stabilizing effect of methanol extract or ethanol extract, EVOO and CPSO were supplemented with synthetic antioxidants TBHQ and BHT at 200 μg/g. A control group was prepared without antioxidants. The flasks were left open and placed in an oil bath at 160 °C to simulate frying. Two samples from each category were removed from the flasks every 4 h for duplicate analysis. The oxidative stability of the oils was evaluated by measurement of the free acidity (percentage of oleic acid), peroxide values (milliequivalents of O_2_/kg oil), and ultraviolet absorption at 232 and 270 nm.

### 3.11. Cell Viability Assay

Cell viability was assayed using the TM_3_ mouse cell line via the MTT assay, as previously described [52]; the specific experimental steps are as follows.

Briefly, 100 µL of a cell suspension (8 × 10^4^ cells/mL) was seeded onto a microplate and incubated at 37 °C for 24 h in an atmosphere of CO_2_ in a humidified incubator. After incubation, 100 μL of methanol extract or ethanol extract of PK insect gall was added. Five duplicate wells in each group were set up. Cell-free medium was employed as a blank group for zero adjustment during measurement. After incubation for 24 h and 48 h, respectively, 20 μL of MTT solution (5 mg/mL) was added and incubated at 37 °C for 4 h. The culture medium was removed, and 200 μL of DMSO was added to solubilize the formazan formed. The absorbance was recorded at 570 nm.

### 3.12. Statistical Analysis

One-way analysis of variance with the post hoc LSD test was conducted to determine significant differences in the total active constituents and biological activity assays between extracts (*p* < 0.05). Exploratory multivariate analysis was performed to cluster extracts, and Pearson’s correlation analysis was completed to assess the relationships between the biological activities and the total active constituents.

## 4. Conclusions

The insect gall of PK has received little scientific attention; therefore, its chemical constituents, biological activities, and safety profile are unknown. This research fills some of these knowledge gaps. The outcomes of phytochemical analysis showed that the insect gall of PK contains 11 types of phytochemicals. the extracts of PK insect gall were prepared using seven solvents. The methanol and ethanol extracts had the highest antioxidant activity. The acetone extract exhibited high antigout and antihypoglycemic activity. The ethyl acetate extract was an excellent urease inhibitor. The dichloromethane extract exhibited inhibitory activity toward tyrosinase. In total, 25 specific compounds were identified from the methanol extract. The methanol and ethanol extracts exhibited excellent stability and safety. They could be used to stabilize olive and sunflower seed oils.

## Figures and Tables

**Figure 1 molecules-28-06021-f001:**
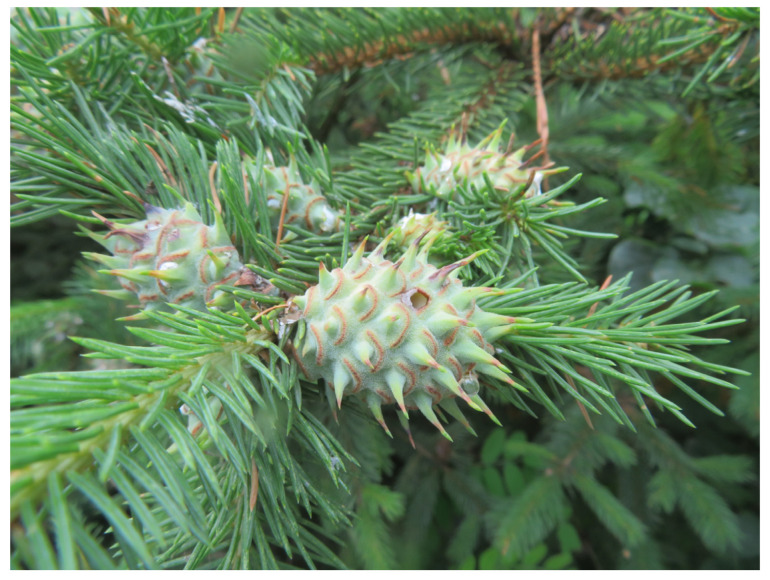
Morphology of insect gall of *Picea koraiensis*.

**Figure 2 molecules-28-06021-f002:**
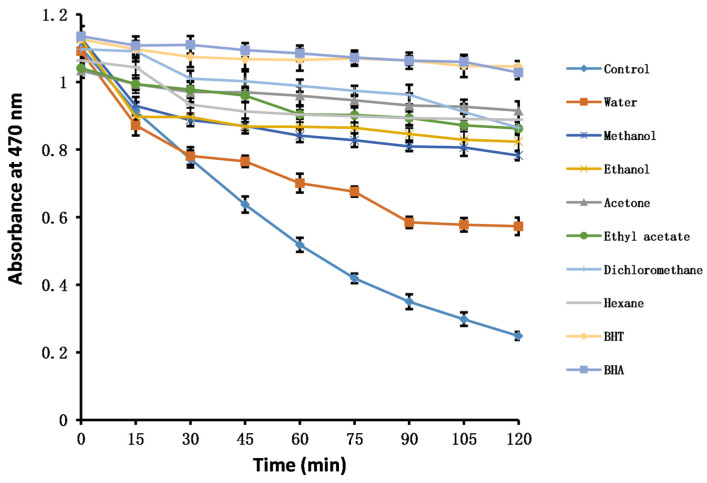
Changes in the absorbance of *β*-carotene in the presence of solvent extracts of insect gall of *Picea koraiensis*. BHT: Butylated hydroxytoluene; BHA: Butyl hydroxyanisole.

**Figure 3 molecules-28-06021-f003:**
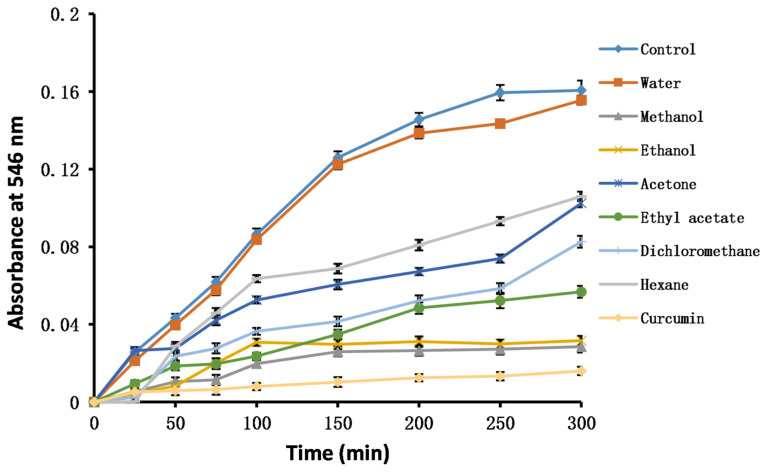
Changes in the absorbance of solvent extracts of insect gall of *Picea koraiensis* over time measured using the nitric oxide scavenging method.

**Figure 4 molecules-28-06021-f004:**
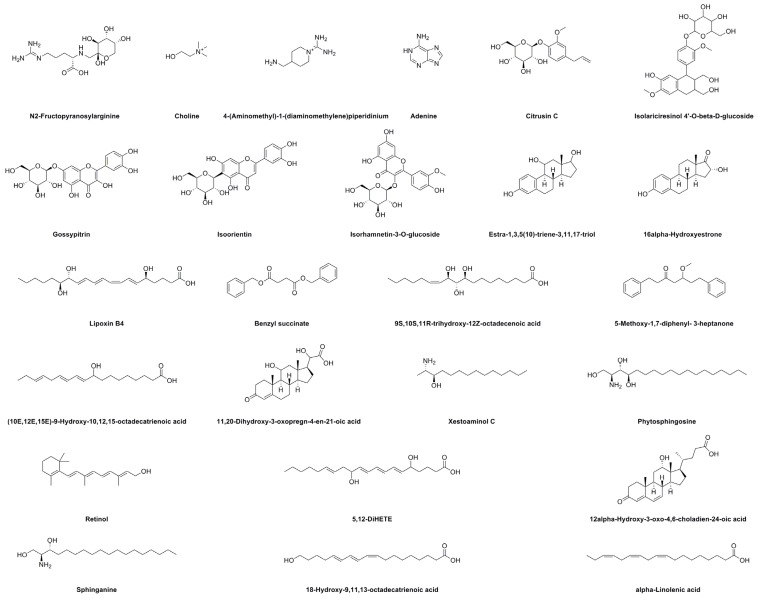
Chemical structures of the compounds identified in methanol extract of insect gall of *Picea koraiensis*.

**Figure 5 molecules-28-06021-f005:**
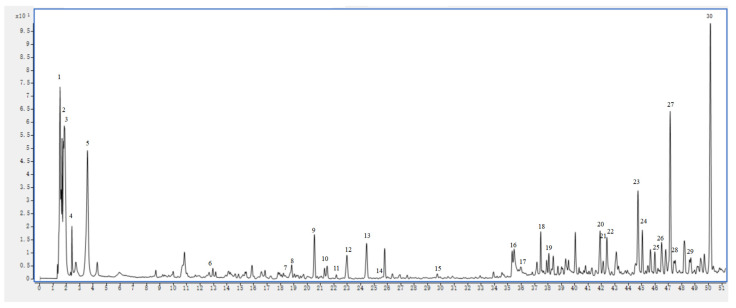
UHPLC–MS results captured in positive-ion mode for methanol extract of insect gall of *Picea koraiensis*.

**Figure 6 molecules-28-06021-f006:**
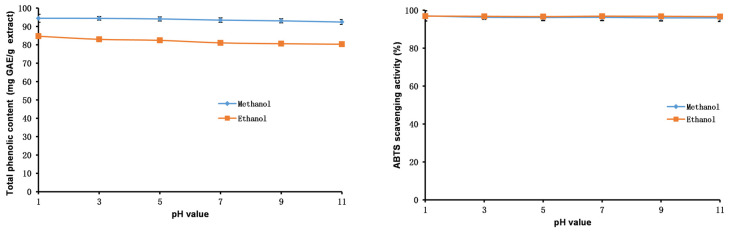
Total phenolic content and ABTS assays to assess the stability of the methanol and ethanol extracts of insect gall of *Picea koraiensis* at various pH values.

**Figure 7 molecules-28-06021-f007:**
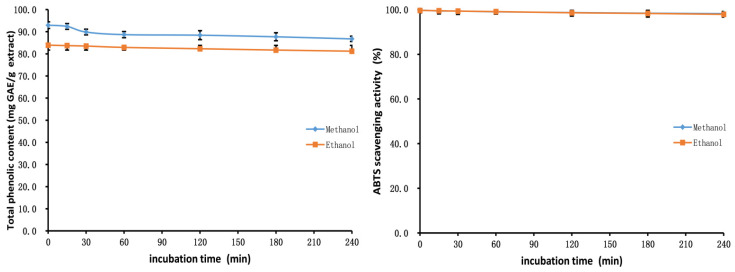
Total phenolic content and ABTS assays to assess the thermal stability of the methanol and ethanol extracts of insect gall of *Picea koraiensis*.

**Figure 8 molecules-28-06021-f008:**
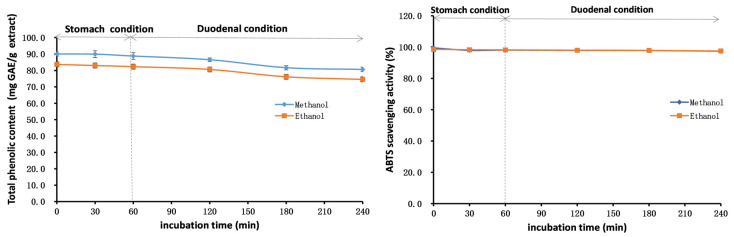
Total phenolic content and ABTS assays to assess the stability of the methanol and ethanol extracts of insect gall of *Picea koraiensis* using an in vitro simulation of the human digestive system.

**Figure 9 molecules-28-06021-f009:**
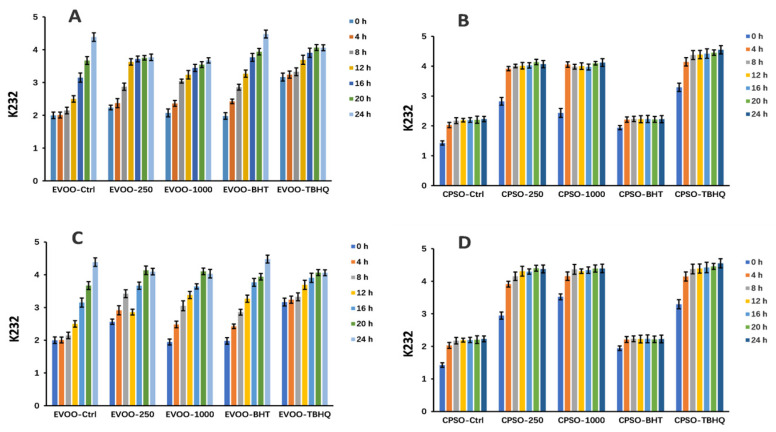
Changes in K_232_ values over time in extra virgin olive oil (EVOO; (**A**,**C**)) and cold-pressed sunflower oil (CPSO; (**B**,**D**)) supplemented with BHT, TBHQ, and two doses of methanol and ethanol extracts of insect gall of *Picea koraiensis*.

**Figure 10 molecules-28-06021-f010:**
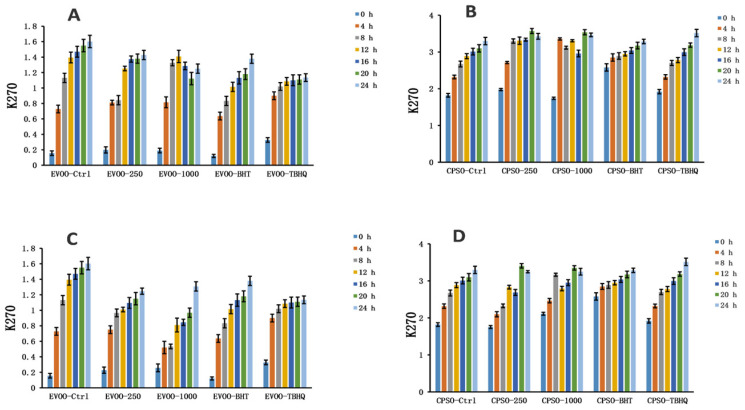
Changes in K_270_ values over time in extra virgin olive oil (EVOO; (**A**,**C**)) and cold-pressed sunflower oil (CPSO; (**B**,**D**)) supplemented with BHT, TBHQ, and two doses of methanol and ethanol extracts of insect gall of *Picea koraiensis*.

**Figure 11 molecules-28-06021-f011:**
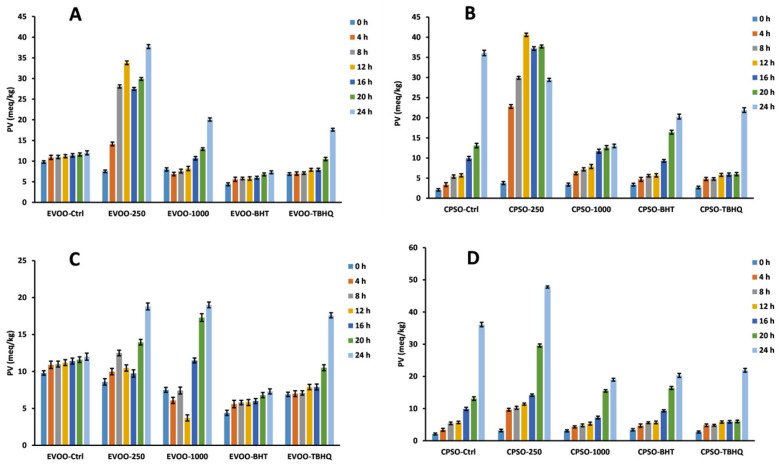
Changes in peroxide values over time in extra virgin olive oil (EVOO; (**A**,**C**)) and cold-pressed sunflower oil (CPSO; (**B**,**D**)) supplemented with BHT, TBHQ, and two doses of methanol and ethanol extracts of insect gall of *Picea koraiensis*.

**Figure 12 molecules-28-06021-f012:**
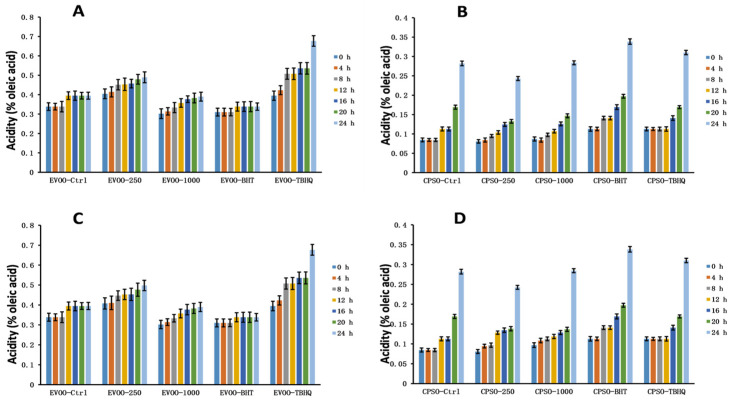
Changes in acidity values over time in extra virgin olive oil (EVOO; (**A**,**C**)) and cold-pressed sunflower oil (CPSO; (**B**,**D**)) supplemented with BHT, TBHQ, and two doses of methanol and ethanol extracts of insect gall of *Picea koraiensis*.

**Figure 13 molecules-28-06021-f013:**
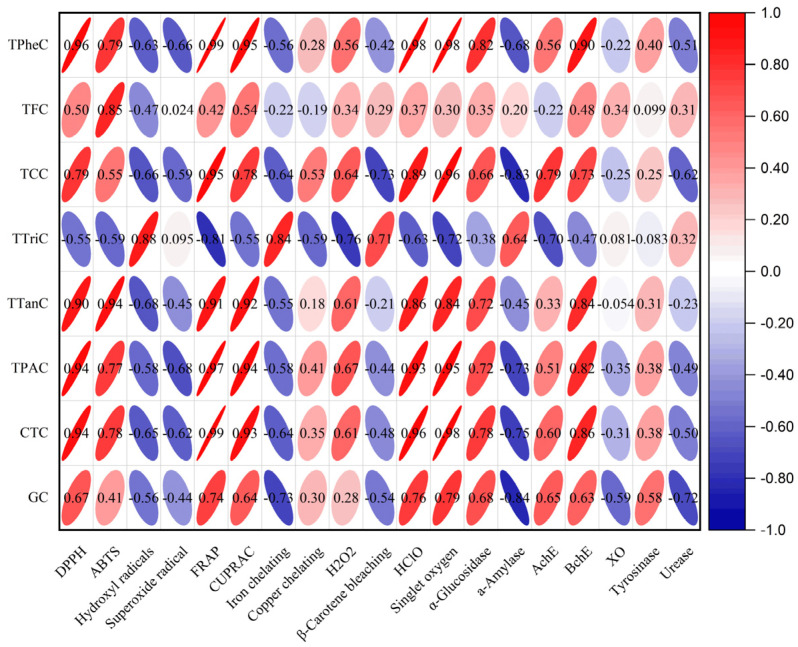
Pearson’s correlation between the biological activities and the total active constituents.

**Figure 14 molecules-28-06021-f014:**
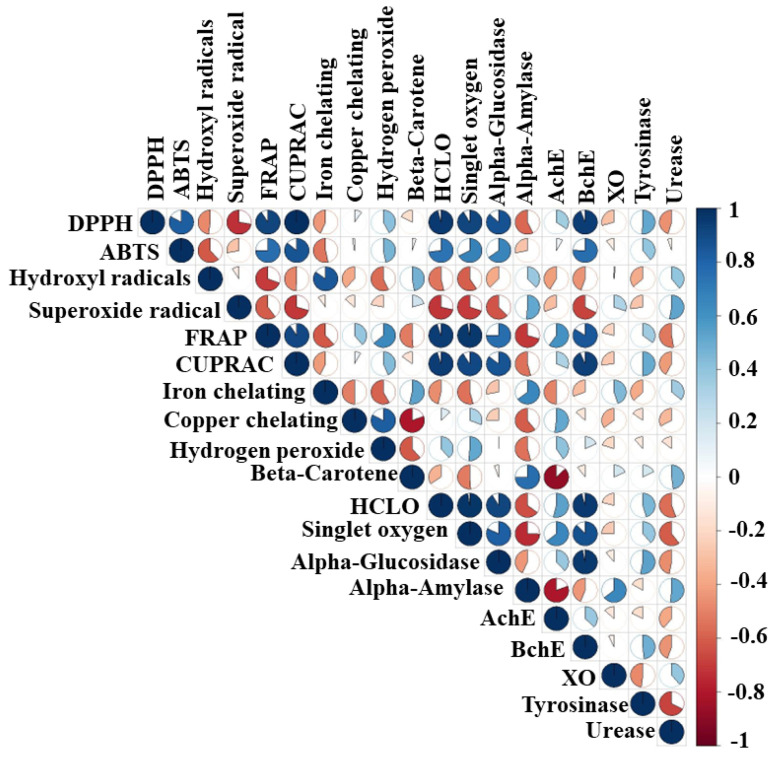
Pearson’s correlations between the biological activities.

**Figure 15 molecules-28-06021-f015:**
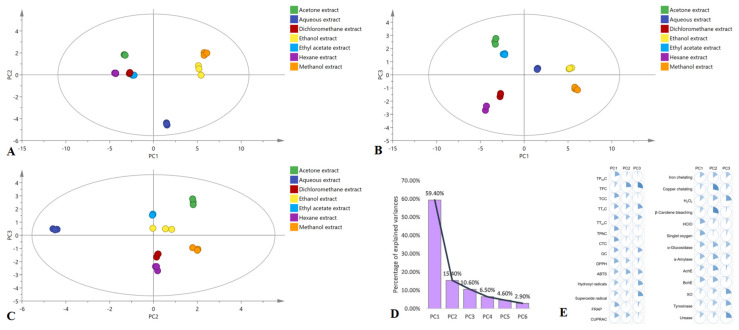
Exploratory multivariate analysis with principal component analysis. (**A**) PC1 vs. PC2; (**B**) PC1 vs. PC3; and (**C**) PC2 vs. PC3. (**D**) Difference explained through each principal component. (**E**) Contribution of biological activities to the principal component construction.

**Figure 16 molecules-28-06021-f016:**
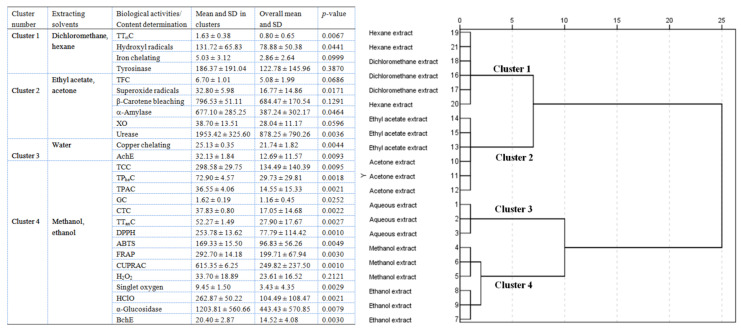
Hierarchical clustering analysis (Euclidean distance, Ward linkage) using the outcomes of the principal component analysis. Only the first three principal components were considered.

**Table 1 molecules-28-06021-t001:** Phytochemical analysis of insect gall of *Picea koraiensis*.

Phytochemicals	Type of Tests	Sample Solution		
		Water	Methanol	Petroleum Ether
Proteins/amino acids	1. Ninhydrin tests	+	○	○
	2. Biuret tests	−	○	○
Carbohydrates	1. Fehling’s tests	+	○	○
	2. Benedict’s tests	+	○	○
	3. Molisch’s tests	+	○	○
	4. Iodine tests	+	○	○
Phenolics	1. FeCl_3_ tests	+	○	○
	2. FeCl_3_-K_3_[Fe(CN)_6_] tests	+	○	○
	3. Diazotization tests	+	○	○
Organic acids	1. pH tests	+	○	○
	2. Blue litmus paper tests	+	○	○
	3. Bromocresol green tests	+	○	○
Tannins	1. FeCl_3_ tests	+	○	○
	2. Bromine water tests	+	○	○
	3. Lead acetate tests	+	○	○
	4. Lime water tests	+	○	○
	5. Gelatin tests	+	○	○
Flavonoids	1. Shinoda tests	○	+	○
	2. Alkaline reagent tests	○	+	○
	3. AlCl_3_ tests	○	+	○
	4. Lead acetate tests	○	+	○
Saponins	1. Foam tests	−	○	○
Steroids and triterpenoids	1. Liebermann–Burchard tests	○	+	○
	2. Salkowski tests	○	+	○
Terpenoids	1. CHCl_3_-H_2_SO_4_ tests	○	+	○
	2. Vanillin-H_2_SO_4_ tests	○	○	+
Alkaloids	1. Bertrad’s reagent tests	○	+	○
	2. Dragendorff’s reagent tests	○	+	○
	3. Mayer’s reagent tests	○	+	○
Anthraquinones	1. Borntrager’s tests	○	−	○
	2. Magnesium acetate tests	○	−	○
Coumarins and lactones	1. Hydroxamic acid iron tests	○	+	○
	2. Diazotization tests	○	+	○
	3. Fluorescence tests	○	+	○
Volatile oils and fats	1. Phosphomolybdic acid tests	○	+	○
	2. Vanillin-H_2_SO_4_ tests	○	+	○
	3. Sudan tests	○	+	○
Cardiac glycosides	1. Kedde tests	○	−	○
	2. Raymond tests	○	−	○
	3. Legal tests	○	−	○
Cyanogenic glycosides	1. Prussian blue tests	−	○	○

(+) indicates presence; (−) indicates absence; (○) indicates no test.

**Table 2 molecules-28-06021-t002:** Extraction yields of insect gall of *Picea koraiensis* extracted with various solvents.

Extracting Solvents	Yields (%, *w*/*w*)
Water	34.32 ± 0.41 ^a^
Methanol	30.45 ± 0.53 ^b^
Ethanol	25.21 ± 1.60 ^b^
Acetone	10.43 ± 1.20 ^c^
Ethyl acetate	8.74 ± 0.79 ^d^
Dichloromethane	7.13 ± 0.64 ^d^
Hexane	5.28 ± 0.21 ^e^

^a–e^ Columns with different superscripts indicate a significant difference (*p* < 0.05). The yield was calculated as % yield = (weight of extract/initial weight of dry sample) × 100.

**Table 3 molecules-28-06021-t003:** Total carbohydrate content (TCC), total protein content (TP_ro_C), total triterpenoid content (TT_ri_C), total phenolic content (TP_he_C), total flavonoid content (TFC), total phenolic acid content (TPAC), gallotannin content (GC), condensed tannin content (CTC), and total tannin content (TT_an_C) of insect gall of *Picea koraiensis* extracted with different solvents.

Extracting Solvents	TCC (mg GE/g Extract)	TP_ro_C (mg BSAE/g Extract)	TT_ri_C (mg GRE/g Extract)	TP_he_C (mg GAE/g Extract)	TFC (mg QE/g Extract)	TPAC (mg CAE/g Extract)	GC (mg GAE/g Extract)	CTC (mg GAE/g Extract)	TT_an_C (mg TAE/g Extract)
Water	277.03 ± 1.61 ^b^	313.32 ± 11.61 ^a^	NONE	34.21 ± 0.52 ^c^	2.92 ± 0.02 ^e^	17.61 ± 0.83 ^c^	1.42 ± 0.13 ^b^	21.21 ± 1.61 ^c^	23.02 ± 0.21 ^e^
Methanol	325.83 ± 0.47 ^a^	NONE	0.31 ± 0.01 ^e^	77.11 ± 0.52 ^a^	6.22 ± 0.12 ^b^	33.22 ± 1.43 ^b^	1.81 ± 0.22 ^a^	38.54 ± 0.23 ^a^	50.91 ± 0.52 ^b^
Ethanol	271.52 ± 1.21 ^c^	NONE	0.29 ± 0.03 ^e^	68.79 ± 1.19 ^b^	7.14 ± 0.23 ^a^	39.92 ± 2.54 ^a^	1.52 ± 0.11 ^b^	37.12 ± 0.22 ^b^	53.64 ± 0.22 ^a^
Acetone	17.02 ± 0.18 ^e^	NONE	1.02 ± 0.01 ^c^	12.52 ± 0.02 ^d^	7.52 ± 0.61 ^a^	3.61 ± 0.32 ^d^	0.36 ± 0.01 ^e^	5.61 ± 0.38 ^e^	28.74 ± 0.47 ^c^
Ethyl acetate	32.14 ± 1.52 ^d^	NONE	0.73 ± 0.02 ^d^	10.31 ± 0.21 ^e^	5.88 ± 0.42 ^b^	4.64 ± 0.32 ^d^	1.02 ± 0.02 ^d^	10.04 ± 0.81 ^d^	24.05 ± 0.21 ^d^
Dichloromethane	12.87 ± 0.22 ^f^	NONE	1.31 ± 0.01 ^b^	4.54 ± 0.12 ^f^	3.57 ± 0.23 ^c^	1.38 ± 0.21 ^e^	1.34 ± 0.11 ^c^	4.73 ± 0.44 ^e^	10.71 ± 0.32 ^f^
Hexane	5.42 ± 0.31 ^g^	NONE	1.89 ± 0.10 ^a^	0.92 ± 0.03 ^g^	2.42 ± 0.22 ^d^	1.49 ± 0.10 ^e^	0.84 ± 0.01 ^f^	2.22 ± 0.23 ^f^	4.32 ± 0.01 ^g^

^a–g^ Columns with different superscripts indicate a significant difference (*p* < 0.05). GE: Glucose equivalent. BSAE: BSA equivalent. GRE: Ginsenoside Re equivalent. GAE: Gallic acid equivalent. QE: Quercetin equivalent. CAE: Caffeic acid equivalent. TAE: Tannic acid equivalent. Values are the mean ± standard deviation of three independent experiments.

**Table 4 molecules-28-06021-t004:** Determination of antioxidant activity of various solvent extracts of insect gall of *Picea koraiensis* using DPPH, ABTS, hydroxyl, and superoxide radicals.

Extracting Solvents	DPPH (mg TE/g Extract)	ABTS (mg TE/g Extract)	Hydroxyl Radicals (mg TE/g Extract)	Superoxide Radicals (%, 2143 μg/mL)
Water	20.54 ± 0.42 ^e^	48.53 ± 1.3 ^f^	<39.00 ^e^	19.81 ± 1.82 ^d^
Methanol	259.43 ± 16.81 ^b^	156.72 ± 8.4 ^d^	<39.00 ^e^	NONE
Ethanol	248.22 ± 9.32 ^c^	182.02 ± 7.0 ^c^	49.72 ± 0.11 ^e^	NONE
Acetone	6.41 ± 0.43 ^f^	100.81 ± 6.0 ^e^	77.41 ± 5.42 ^d^	27.62 ± 2.22 ^c^
Ethyl acetate	7.74 ± 0.42 ^f^	110.72 ± 2.1 ^e^	83.62 ± 7.61 ^d^	38.12 ± 1.61 ^b^
Dichloromethane	2.03 ± 0.22 ^f^	62.43 ± 2.0 ^f^	72.61 ± 6.42 ^d^	28.71 ± 1.52 ^c^
Hexane	<0.44 ^f^	16.71 ± 0.0 ^g^	190.81 ± 17.63 ^c^	3.31 ± 0.21 ^e^
*L*-ascorbic acid *	1111.13 ± 8.01 ^a^	1118.30 ± 32.89 ^a^	1119.43 ± 3.91 ^a^	N.T.
BHT *	215.12 ± 2.32 ^d^	808.51 ± 10.32 ^b^	468.81 ± 6.31 ^b^	N.T.
Curcumin *	N.T.	N.T.	N.T.	45.62 ± 1.01 ^a^

^a–g^ Columns with different superscripts indicate a significant difference (*p* < 0.05). * Used as a standard antioxidant. N.T. indicates no test. TE: Trolox equivalent. BHT: Butylated hydroxytoluene. DPPH: 2,2-Diphenyl-1-picrylhydrazyl. ABTS: 2,2′-Azino-bis(3-ethylbenzothiazoline-6-sulphonic acid) diammonium salt. Values are the mean ± standard deviation of three independent experiments.

**Table 5 molecules-28-06021-t005:** Determination of antioxidant activity of various solvent extracts of insect gall of *Picea koraiensis* using FRAP, CUPRAC, and metal chelating.

Extracting Solvents	FRAP (mg TE/g Extract)	CUPRAC (mg TE/g Extract)	Iron Chelating (mg EDTAE/g Extract)	Copper Chelating (mg EDTAE/g Extract)
Water	228.02 ± 1.62 ^c^	118.61 ± 8.31 ^d^	0.52 ± 0.02 ^e^	25.12 ± 0.42 ^a^
Methanol	298.31 ± 17.43 ^b^	615.33 ± 9.41 ^c^	1.89 ± 0.01 ^c^	20.31 ± 0.51 ^c^
Ethanol	287.02 ± 10.24 ^b^	615.27 ± 3.02 ^c^	0.51 ± 0.02 ^e^	23.45 ± 1.41 ^b^
Acetone	162.84 ± 1.81 ^d^	126.89 ± 2.81 ^d^	5.56 ± 0.13 ^b^	<20.80 ^c^
Ethyl acetate	155.02 ± 1.52 ^d^	108.32 ±7.67 ^e^	1.52 ± 0.11 ^d^	<20.80 ^c^
Dichloromethane	142.02 ± 5.14 ^d^	87.41 ± 4.02 ^f^	2.17 ± 0.21 ^c^	<20.80 ^c^
Hexane	124.78 ± 2.22 ^e^	76.78 ± 3.42 ^f^	7.88 ± 0.37 ^a^	<20.80 ^c^
*L*-ascorbic acid *	980.03 ± 10.02 ^a^	1401.56 ± 10.32 ^b^	N.T.	N.T.
BHT *	310.02 ± 0.81 ^b^	1530.01 ± 11.41 ^a^	N.T.	N.T.

^a–f^ Columns with different superscripts indicate a significant difference (*p* < 0.05). * Used as a standard antioxidant. N.T. indicates no test. FRAP: Ferric-reducing antioxidant power. CUPRAC: Cupric ion reducing antioxidant capacity. BHT: Butylated hydroxytoluene. TE: Trolox equivalent. EDTAE: EDTA equivalent. Values are the mean ± standard deviation of three independent experiments.

**Table 6 molecules-28-06021-t006:** Determination of antioxidant activity of various solvent extracts of insect gall of *Picea koraiensis* using H_2_O_2_, *β*-carotene bleaching, singlet oxygen, and HClO.

Extracting Solvents	H_2_O_2_ (mg FAE/g Extract)	*β*-Carotene Bleaching AAC	Singlet Oxygen (%, 2000 μg/mL)	HClO (mg LAE/g Extract)
Water	42.89 ± 0.11 ^c^	312.71 ± 2.00 ^d^	5.12 ± 0.21 ^d^	97.72 ± 0.13 ^c^
Methanol	16.62 ± 1.42 ^e^	643.32 ± 10.74 ^c^	10.61 ± 1.12 ^b^	307.25 ± 18.22 ^a^
Ethanol	50.81 ± 3.61 ^b^	659.33 ± 9.03 ^c^	8.32 ± 0.84 ^c^	218.52 ± 8.81 ^b^
Acetone	24.42 ± 1.23 ^d^	823.32 ± 12.61 ^b^	NONE	<27 ^d^
Ethyl acetate	18.45 ± 1.81 ^e^	769.82 ± 21.62 ^b^	NONE	<27 ^d^
Dichloromethane	<6.00 ^f^	807.24 ± 21.53 ^b^	NONE	<27 ^d^
Hexane	<6.00 ^f^	775.73 ± 11.20 ^b^	NONE	<27 ^d^
BHT *	N.T.	876.94 ± 4.61 ^a^	N.T.	N.T.
BHA *	N.T.	883.03 ± 3.42 ^a^	N.T.	N.T.
Ferulic acid *	1000.03 ± 12.77 ^a^	N.T.	90.32 ± 1.21 ^a^	N.T.

^a–f^ Columns with different superscripts indicate a significant difference (*p* < 0.05). * Used as a standard antioxidant. N.T. indicates no test. H_2_O_2_: Hydrogen peroxide. HClO: Hypochlorous acid. AAC: Antioxidant activity coefficient. FAE: Ferulic acid equivalent. LAE: Lipoic acid equivalent. Values are the mean ± standard deviation of three independent experiments.

**Table 7 molecules-28-06021-t007:** Determination of the enzyme-inhibitory activity of various solvent extracts of insect gall of *Picea koraiensis* toward *α*-glucosidase, *α*-amylase, AChE, BChE, XO, tyrosinase, and urease.

Extracting Solvents	*α*-Glucosidase (mg AE/g Extract)	*α*-Amylase (mg AE/g Extract)	AChE (100 μg/mL)	BChE (mg DE/g Extract)	XO (1250 μg/mL)	Tyrosinase (mg A_rb_E/g Extract)	Urease (mg T_hi_E/g Extract)
Water	<49.02 ^e^	<36.00 ^g^	32.11 ± 1.81 ^b^	<12.01 ^e^	27.24 ± 1.52 ^d^	<1.01 ^c^	256.90 ± 0.03 ^e^
Methanol	1713.62 ± 78.62 ^a^	173.24 ± 6.02 ^e^	25.42 ± 2.02 ^c^	23.02 ± 0.71 ^b^	32.03 ± 0.54 ^c^	247.41 ± 9.89 ^b^	102.04 ± 8.02 ^e^
Ethanol	694.13 ± 11.64 ^b^	89.78 ± 4.6 ^f^	10.45 ± 0.89 ^e^	17.81 ± 0.63 ^c^	13.41 ± 2.31 ^f^	236.34 ± 13.92 ^b^	583.52 ± 12.24 ^d^
Acetone	<49.02 ^e^	940.03 ± 58.89 ^a^	NONE	12.91 ± 0.22 ^d^	50.78 ± 3.89 ^b^	<1.01 ^c^	1672.16 ± 87.53 ^b^
Ethyl acetate	181.82 ± 3.12 ^d^	416.94 ± 11.56 ^d^	12.43 ± 0.44 ^d^	<12.01 ^e^	26.62 ± 1.32 ^d^	<1.01 ^c^	2234.58 ± 143.24 ^a^
Dichloromethane	240.03 ± 4.82 ^c^	588.43 ± 11.53 ^b^	NONE	<12.01 ^e^	20.51 ± 0.89 ^e^	360.72 ± 5.93 ^a^	152.78 ± 11.67 ^e^
Hexane	176.61 ± 1.93 ^d^	469.12 ± 14.72 ^c^	8.34 ± 0.43 ^f^	<12.01 ^e^	25.61 ± 1.21 ^d^	12.03 ± 0.94 ^c^	1145.81 ± 39.72 ^c^
Donepezil *	N.T.	N.T.	99.81 ± 0.11 ^a^	1000.02 ± 9.21 ^a^	N.T.	N.T.	N.T.
Allopurinol *	N.T.	N.T.	N.T.	N.T.	95.22 ± 0.11 ^a^	N.T.	N.T.

^a–g^ Columns with different superscripts indicate a significant difference (*p* < 0.05). * Used as a positive control. N.T. indicates no test. AChE: Acetylcholinesterase. BChE: Butyrylcholinesterase. XO: Xanthine oxidase. A_rb_E: Arbutin equivalent. T_hi_E: Thiourea equivalent. Values are the mean ± standard deviation of three independent experiments.

**Table 8 molecules-28-06021-t008:** Compounds identified in methanol extract of insect gall of *Picea koraiensis* by UHPLC–MS.

Peak No.	RT (min)	Identification	Molecular Formula	Selective Ion	Full Scan MS (*m*/*z*)		Error (ppm)	MS/MS Fragments (*m*/*z*)
					Theory	Measured		
1	1.42	N2-Fructopyranosylarginine	C_12_H_24_N_4_O_7_	[M + H]^+^	337.1714	337.1717	−0.9	251.0314
2	1.52	Choline	C_5_H_14_NO^+^	[M]^+^	104.1070	104.1069	1.0	—
3	1.85	4-(Aminomethyl)-1-(diaminomethylene)piperidinium	C_7_H_17_N_4_^+^	[M + H]^+^	158.1526	158.1538	−7.6	140.1438
4	2.41	Adenine	C_5_H_5_N_5_	[M + H]^+^	136.0623	136.0621	1.5	118.0859
5	3.58	Unknown				140.1435		98.0964
6	12.66	Citrusin C	C_16_H_22_O_7_	[M + Na]^+^	349.1263	349.1260	0.9	344.1706
7	18.85	Isolariciresinol 4′-*O*-beta-D-glucoside	C_26_H_34_O_11_	[M + Na]^+^	545.1999	545.1999	0	540.2458, 285.1132
8	19.05	Gossypitrin	C_21_H_20_O_12_	[M + H]^+^	465.1033	465.1027	1.3	303.0507
9	20.55	Unknown				369.1537		167.1074
10	21.49	Isoorientin	C_21_H_20_O_11_	[M + H]^+^	449.1084	449.1072	2.7	287.0551
11	22.17	Isorhamnetin-3-*O*-glucoside	C_22_H_22_O_12_	[M + H]^+^	479.1190	479.1180	2.1	317.0665
12	22.97	Estra-1,3,5(10)-triene-3,11,17-triol	C_18_H_24_O_3_	[M + NH_4_]^+^	306.2069	306.2071	−0.7	107.0490
13	24.45	16*α*-Hydroxyestrone	C_18_H_22_O_3_	[M + NH_4_]^+^	304.1913	304.1915	−0.7	112.1111
14	25.79	Unknown				466.2669		335.0948
15	29.74	Lipoxin B4	C_20_H_32_O_5_	[M + Na]^+^	375.2147	375.2142	1.3	309.0986
16	35.48	Benzyl succinate	C_18_H_18_O_4_	[M + H]^+^	299.1298	299.1300	−0.7	91.0419, 77.0386
17	35.96	9*S*,10*S*,11*R*-trihydroxy-12*Z*-octadecenoic acid	C_18_H_34_O_5_	[M + Na]^+^	353.2304	353.2299	1.4	301.1358
18	37.46	5-Methoxy-1,7-diphenyl-3-heptanone	C_20_H_24_O_2_	[M + H]^+^	297.1855	297.1856	−0.3	282.1622
19	38.08	(10*E*,12*E*,15*E*)-9-Hydroxy-10,12,15-octadecatrienoic acid	C_18_H_30_O_3_	[M + Na]^+^	317.2093	317.2117	−7.6	253.1952
20	41.86	11,20-Dihydroxy-3-oxopregn-4-en-21-oic acid	C_21_H_30_O_5_	[M + H]^+^	363.2171	363.2173	−0.6	317.2122
21	42.11	Xestoaminol C	C_14_H_31_NO	[M + H]^+^	230.2484	230.2481	1.3	212.2368, 201.1640
22	42.40	Phytosphingosine	C_18_H_39_NO_3_	[M + H]^+^	318.3008	318.3006	0.6	302.2220, 301.2139
23	44.73	Unknown				301.2165		283.2068, 255.2115
24	45.05	Unknown				385.2381		128.0620
25	46.29	Retinol	C_20_H_30_O	[M + H]^+^	287.2375	287.2371	1.4	269.2265
26	46.49	5,12-DiHETE	C_20_H_32_O_4_	[M + H]^+^	337.2379	337.2362	5.0	279.2330
27	47.13	12*α*-Hydroxy-3-oxo-4,6-choladien-24-oic acid	C_24_H_34_O_4_	[M + H]^+^	387.2535	387.2533	0.5	269.2280
28	47.48	Sphinganine	C_18_H_39_NO_2_	[M + H]^+^	302.3059	302.3048	3.6	303.3090
29	48.62	18-Hydroxy-9,11,13-octadecatrienoic acid	C_18_H_30_O_3_	[M + Na]^+^	317.2093	317.2110	−5.4	318.2156
30	50.14	*α*-Linolenic acid	C_18_H_30_O_2_	[M + Na]^+^	301.2143	301.2160	−5.6	183.1169, 169.1012

RT: Retention time.

**Table 9 molecules-28-06021-t009:** Determination of cytotoxicity of methanol and ethanol extracts of insect gall of *Picea koraiensis* by the MTT method.

Methanol Extract (μg/mL)	Cell Survival Rate of TM_3_ Cells (%)		Ethanol Extract (μg/mL)	Cell Survival Rate of TM_3_ Cells (%)	
	24 h	48 h		24 h	48 h
0	100.00 ± 0.64 ^d^	100.03 ± 0.73 ^a^	0	100.00 ± 0.52 ^b^	100.00 ± 0.63 ^a^
25	107.58 ± 2.14 ^c^	95.12 ± 1.45 ^b^	25	97.31 ± 1.89 ^c^	96.23 ± 1.56 ^b^
50	101.78 ± 1.67 ^d^	96.78 ± 1.44 ^b^	50	114.49 ± 2.74 ^a^	111.02 ± 3.21 ^a^
100	111.80 ± 2.42 ^b^	91.64 ± 1.89 ^c^	100	103.52 ± 2.43 ^b^	111.72 ± 2.56 ^a^
200	116.42 ± 2.81 ^a^	96.71 ± 2.33 ^b^	200	101.53 ± 1.42 ^b^	108.89 ± 1.63 ^a^

^a–d^ Columns with different superscripts indicate a significant difference (*p* < 0.05). Values are the mean ± standard error of five independent experiments.

## Data Availability

All data presented in this study are available in the article.

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
