# Peer review of "Phytochemical Analysis, Antioxidant and Enzyme-Inhibitory Activities, and Multivariate Analysis of Insect Gall Extracts of *Picea koraiensis* Nakai"

_molecules, 2023, doi:10.3390/molecules28166021_

Round 1

Reviewer 1 Report (New Reviewer)

This work is interesting and matching with the world's needs nowadays, they need natural materials to supplement functional foods or preserve them without side effects.  moreover, I have a few comments:

1. you need to recheck spaces and punctuation in all paper parts

2. the materials and methods parts were unacceptable and need to be rewritten with detail (all methods are self-citations, which is unacceptable), and several tests were not maintained and I found it in the results which made using and utilizing this paper hard on other scientists. 

3. You mentioned that the identification of specimens was done by Professor  Junlin but you didn't mention what is the specific field of him.

4. you also said that you made an extract with different 7 solvents without any details about the condition of extraction, how is that?

5. this article was full of abbreviations which need to identify first as appendix 

6. recheck results and discussions, several parts had discussions without reference to confirm your point of view

7. there is no need for several figures such as No. 2 and 4 

8. you need to rearrange the part of the results because I found some parts were far away from its tables, such as quantitive phytochemicals of TTC and TPC .. etc    

9. what did you mean by N.T also what does no test mean? 

10. Are carbohydrates considered as phytochemical components, could you explain that?

Minor editing of English language required

Author Response

Question 1: This work is interesting and matching with the world's needs nowadays, they need natural materials to supplement functional foods or preserve them without side effects.  moreover, I have a few comments:

Answer: Dear reviewer, thank you very much for your recognition of our work. And thank you for your valuable comments to help us improve our manuscript.

Question 2: you need to recheck spaces and punctuation in all paper parts

Answer: Thank you for your reminder! We have rechecked spaces and punctuation in all paper parts.

Question 3: the materials and methods parts were unacceptable and need to be rewritten with detail (all methods are self-citations, which is unacceptable), and several tests were not maintained and I found it in the results which made using and utilizing this paper hard on other scientists. 

Answer: Your suggestion is very good. After revising according to your requirements, the content is more complete. This allows other scientists to use and utilize this paper well.

Question 4: You mentioned that the identification of specimens was done by Professor  Junlin but you didn't mention what is the specific field of him.

Answer: Professor  Junlin is a plant taxonomist. We have added this information to the manuscript.

Question 5: you also said that you made an extract with different 7 solvents without any details about the condition of extraction, how is that?

Answer: We are very sorry that we forgot to add this content to the manuscript. It has now been added to the section 3.3.

Question 6: this article was full of abbreviations which need to identify first as appendix

Answer: What you said is very correct! We have removed unnecessary abbreviations and retained the full name to avoid misunderstandings among readers.

Question 7: recheck results and discussions, several parts had discussions without reference to confirm your point of view

Answer: Dear reviewer, we have rechecked according to your request and correctly cited.

Question 8: there is no need for several figures such as No. 2 and 4 

Answer: Figure 2 shows the chemical structures of the compounds identified in methanol extract of Picea koraiensis insect gall. Figure 4 shows the changes in the absorbance of β-carotene in the presence of solvent extracts of PK insect gall. I think they need to be reserved. This is because the existence of Figure 4 could help readers directly see the structure of the compounds without query. The existence of Figure 2 could help readers more directly see the changes in the absorbance of β-carotene. Although there is also a description of β-carotene in Table 7, it can only see the final results and cannot see the changes in absorbance values during the experimental process. But if you think they must be deleted, I will put them in the Supplementary Material.

Question 9: you need to rearrange the part of the results because I found some parts were far away from its tables, such as quantitive phytochemicals of TTC and TPC .. etc   

Answer: Your suggestion is very good! We have rearranged the part of the results.

Question 10: what did you mean by N.T. also what does no test mean?

Answer: N.T. is the abbreviation for no test. For example, in table 5, in the superoxide radials experiment, a standard antioxidant curcumin was used. However, L-ascorbic acid and BHT were not positive controls for this experiment, so their superoxide radial scavenging ability was not tested.

Question 11: Are carbohydrates considered as phytochemical components, could you explain that?

Answer: Dear reviewer, I'm sorry to have caused you confusion. Let me explain below. The carbohydrates here not only include sugars, but also glycosides. This detection method can detect both of them. The results of aqueous extracts are mainly composed of sugars, while the results of other solvent extracts are mainly composed of glycosides.

Reviewer 2 Report (New Reviewer)

Introduction - Instead of the overly long description of the PK galls, there should be information presented on phytochemical and biological studies on P. koraiensis cones (and galls) conducted in recent years by various researchers (for example, Jia et al. Front. Plant Sci. 2020, 3011:580155; Latos-Brozio et al. Antioxidants 2021, 10, 1723 and other) as these investigations indicate the presence of similar groups of compounds (mainly polyphenols) that are usually produced in cones as well as in galls in response to insect-induced tissue damage.

2.1. I don't see the point of performing and presenting the results of general qualitative tests for the groups of phytoconstituents, especially since they were basically performed only for the aqueous extract. After all, the authors used very sophisticated tools for the qualitative (and quantitative) analysis – UPLC coupled with an orbitrap mass spectrometer. Table 1 and the whole description should be removed from the text.

2.3. and Table 3 - I have some objections to the reliability of the results of the qualitative analysis presented in Table 3. The exact m/z values for molecular ions (or their derivatives) and major fragment ions do not clearly confirm the structure of compounds (especially phenolics), since there are a large number of position isomers with the same molecular weight. What about analyzing UV spectra for chromophore compounds? Data should be included and compared with the results of other researchers. Why the results of qualitative analysis of other polar extracts (aqueous or ethanolic) were not presented and discussed to confirm or exclude the presence of specific compounds or to complete the data on the qualitative profile of components present in the PK galls? The results of LC/MS analysis for the remaining extracts should be included as TICs (or BPCs) in the Supplementary Materials.

Figure 3 – The quality of the photo should be increased i.e. the upper description, the frame, and the grey background should be removed

2.4. The quantitative analysis of extracts is essentially a semi-quantitative procedure, for which general, low-precision tests were used to indicate the content of certain groups of phytoconstituents. The correct approach in such circumstances is indicated by the monographs for herbal preparations contained in various pharmacopeias around the world, where the identity of the main groups of compounds responsible for biological effects is first determined (the qualitative analysis) and then their content per major component is calculated (the quantitative analysis). The Authors should do this for all the extracts obtained (i.e. identify and quantify the proper bioactive components) so that further biological studies and correlation of results would be reliable. It is possible since you have very good analytical equipment (UPLC coupled with an orbitrap mass spectrometer). I think numerous published data concerning phytochemical studies on P. koraiensis will be also helpful.

2.4.1. It is completely incomprehensible, why GAE (gallic acid equivalent) was used to calculate total carbohydrate content in extracts while this indicator allows quantifying the total phenolic content (TPC) in herbal preparations. The authors said this might be because many of the components of the methanol extract were glycosides. This statement is false, as only 5 compounds (6-8 and 10-11) out of 30 belong to the glycosides.

2.4.2. What was the purpose of determining total protein content in extracts? Do you want to use the PK galls as food?

2.5.1 You wrote that the methanolic and ethanolic extracts of PK insect galls showed strong antioxidant capacity attributed to high isoorientin, citrusin C, and gossypitrin contents. I did not find any quantitative results for these phenolics in the aforementioned extracts and their peaks in Figure 2 are very small.

2.5.2, 2.5.3, 2.6.3, and 2.6.4 - Similar comments as for 2.5.1 – some compounds responsible for the antioxidant or other biological activities are indicated by the Authors but not quantified in individual extracts.

2.5.8. A high level of alpha-linolenic acid (ALA) has not been determined in the methanolic extract of PK insect galls therefore conclusions about NO scavenging activity are unauthorized. The same objection applies to the activity of ALA in AChE and BChE inhibitory assays (2.6.2). Besides, ALA is a strongly hydrophobic compound, which should be present in higher concentrations in non-polar PK gall extracts and determine their stronger activity. The results of ALA content in all extracts are essential.

2.7. What was the aim of the performed stability studies of methanolic and ethanolic extracts? Without knowing the content of the individual bioactive components in these extracts, the results of the studies conducted are pointless and should be removed from the text together with Figures 6-8.

2.8. Could you explain how the methanolic extract of PK galls can be used as an antioxidant (stabilizing) additive for olive and sunflower oils intended for human consumption?

Table 9 is sufficient to show the results of cytotoxicity of methanolic and ethanolic extracts of PK insect galls. Figures 13 and 14 should be removed and included in the Supplementary Materials.

2.10. The authors wrote that: “a good correlation was found between the TCC values and its antioxidant activity (Figure 15). This implies that the carbohydrates in the methanolic extract are flavonoid glycosides such as luteolin-4'-O-glucoside” The problem is that this compound (luteolin-4'-O-glucoside) was not identified at all in the methanolic extract. And of course, it is not a carbohydrate!

My general comments are that a detailed qualitative and quantitative analysis for individual PK gall extracts should be done and the results presented. This is a prerequisite for further proper evaluation of the biological properties of the extracts and multivariate statistical analysis. Major correction and further peer-review evaluation of the obtained research results is essential.

Minor correction of the text is needed.

Author Response

Question 1: Introduction - Instead of the overly long description of the PK galls, there should be information presented on phytochemical and biological studies on P. koraiensis cones (and galls) conducted in recent years by various researchers (for example, Jia et al. Front. Plant Sci. 2020, 3011:580155; Latos-Brozio et al. Antioxidants 2021, 10, 1723 and other) as these investigations indicate the presence of similar groups of compounds (mainly polyphenols) that are usually produced in cones as well as in galls in response to insect-induced tissue damage.

Answer: Dear reviewer, thank you very much for your recognition of our work. And thank you for your valuable comments to help us improve our manuscript. Your suggestion is very good. After revising according to your requirements, the content is more complete. It could make readers have a better understanding on phytochemical and biological studies on P. koraiensis cones (and galls). Thank you very much!

Question 2: 2.1. I don't see the point of performing and presenting the results of general qualitative tests for the groups of phytoconstituents, especially since they were basically performed only for the aqueous extract. After all, the authors used very sophisticated tools for the qualitative (and quantitative) analysis – UPLC coupled with an orbitrap mass spectrometer. Table 1 and the whole description should be removed from the text.

Answer: Your question is very professional! Let me explain to you below. Our overall design approach is to preliminarily determine which phytochemicals are present in plants through qualitative experiments. As you mentioned, in general, this result does not have much significance as there may be false positive or false negative results. Therefore, we considered these issues when designing qualitative experiments. The qualitative experiments for each type of phytochemical include at least two experiments, with a maximum of five experiments. The purpose is to hope that they can verify each other and avoid drawing incorrect conclusions. Because we continue to conduct quantitative experiments based on the results of qualitative experiments. Quantitative analysis of such phytochemicals is only performed when the results of qualitative experiments are positive. Therefore, this is also the reason why the results of qualitative experiments are important in our experiments.

It also needs to be explained that our qualitative experiment is not only for aqueous extraction solution, but also for methanol extraction solution and petroleum ether extraction solution. Aqueous extraction solution is used to check for carbohydrates, organic acids, saponins, glycosides, phenols, tannins and cyanogenic glycosides.These types of phytochemicals have high polarity and could be dissolved by water. Methanol extraction solution is used to check for flavonoids, anthraquinones, cardiac glycosides, coumarins, lactones, volatile oils, terpenoids, steroids and lipids.These types of phytochemicals have relative high polarity and could be dissolved by methanol. Petroleum ether extraction solution is used to check for volatile oils and lipids, steroids or triterpenoids. These types of phytochemicals have low polarity and could be dissolved by petroleum ether. Although the phytochemicals contained in the three extraction solution will intersect, however, in general, once a positive result was shown in one extraction solution, the other two extraction solutions do not have to be. In summary, we would like to retain Table 1 and the whole description.

Question 3: 2.3. and Table 3 - I have some objections to the reliability of the results of the qualitative analysis presented in Table 3. The exact m/z values for molecular ions (or their derivatives) and major fragment ions do not clearly confirm the structure of compounds (especially phenolics), since there are a large number of position isomers with the same molecular weight. What about analyzing UV spectra for chromophore compounds? Data should be included and compared with the results of other researchers. Why the results of qualitative analysis of other polar extracts (aqueous or ethanolic) were not presented and discussed to confirm or exclude the presence of specific compounds or to complete the data on the qualitative profile of components present in the PK galls? The results of LC/MS analysis for the remaining extracts should be included as TICs (or BPCs) in the Supplementary Materials.

Answer: Your question is very normal. Indeed, as you mentioned, there is isomerism in phenolic compounds, However, based on our experimental results, a total of four phenolic compounds (two flavonoid glycosides and two lignin) were identified, their isomers are relatively few, and we have carefully compared them with literature, which shows high reliability.  The logical order of the article writing is incorrect. It should be through qualitative and quantitative experiments and biological activity experiments that we have identified methanol extract as the object of further research. Therefore, no UHPLC-MS analysis was performed on other extracts.

Question 4: Figure 3 – The quality of the photo should be increased i.e. the upper description, the frame, and the grey background should be removed

Answer: The upper description have been removed. But the frame and grey background could not be removed. Because the image is exported from the software, and they cannot be removed.

Question 5: 2.4. The quantitative analysis of extracts is essentially a semi-quantitative procedure, for which general, low-precision tests were used to indicate the content of certain groups of phytoconstituents. The correct approach in such circumstances is indicated by the monographs for herbal preparations contained in various pharmacopeias around the world, where the identity of the main groups of compounds responsible for biological effects is first determined (the qualitative analysis) and then their content per major component is calculated (the quantitative analysis). The Authors should do this for all the extracts obtained (i.e. identify and quantify the proper bioactive components) so that further biological studies and correlation of results would be reliable. It is possible since you have very good analytical equipment (UPLC coupled with an orbitrap mass spectrometer). I think numerous published data concerning phytochemical studies on P. koraiensis will be also helpful.

Answer: Your suggestion is very good! But our current experimental conditions are unable to complete it. On the one hand, because the analytical equipment is jointly completed by our collaborators, rather than owned by ourselves, it involves a large amount of costs. On the other hand, there are not many reports on the composition research of PK galls, and we have also presented more comprehensive information about it through our qualitative and quantitative experiments, Moreover, it still has a long way to go before it can be turned into a drug, so this study has not yet used a pharmacopoeia to require it. Your suggestions on adopting the data concerning phytochemical studies on P. koraiensis are very good!

Question 6: 2.4.1. It is completely incomprehensible, why GAE (gallic acid equivalent) was used to calculate total carbohydrate content in extracts while this indicator allows quantifying the total phenolic content (TPC) in herbal preparations. The authors said this might be because many of the components of the methanol extract were glycosides. This statement is false, as only 5 compounds (6-8 and 10-11) out of 30 belong to the glycosides.

Answer: I'm very sorry. There is a mistake in writing here. The correct one should be GE (glucose equivalent). The false statement also corrected.

Question 7: 2.4.2. What was the purpose of determining total protein content in extracts? Do you want to use the PK galls as food?

Answer: We provided an explanation in response to question 2. Quantitative analysis of such phytochemicals is only performed when the results of qualitative experiments are positive. Due to the positive results in the qualitative experiment of protein/amino acids, we conducted quantitative experiments on them. We think that use the PK galls as food still requires a long way to go.

Question 8: 2.5.1 You wrote that the methanolic and ethanolic extracts of PK insect galls showed strong antioxidant capacity attributed to high isoorientin, citrusin C, and gossypitrin contents. I did not find any quantitative results for these phenolics in the aforementioned extracts and their peaks in Figure 2 are very small.

Answer: Thank you very much for your reminder! Our statements are not rigorous and have been modified.

Question 9: 2.5.2, 2.5.3, 2.6.3, and 2.6.4 - Similar comments as for 2.5.1 – some compounds responsible for the antioxidant or other biological activities are indicated by the Authors but not quantified in individual extracts.

Answer: Our statements are not rigorous and have been modified.

Question 10: 2.5.8. A high level of alpha-linolenic acid (ALA) has not been determined in the methanolic extract of PK insect galls therefore conclusions about NO scavenging activity are unauthorized. The same objection applies to the activity of ALA in AChE and BChE inhibitory assays (2.6.2). Besides, ALA is a strongly hydrophobic compound, which should be present in higher concentrations in non-polar PK gall extracts and determine their stronger activity. The results of ALA content in all extracts are essential.

Answer: Our statements are not rigorous and have been modified.

Question 11: 2.7. What was the aim of the performed stability studies of methanolic and ethanolic extracts? Without knowing the content of the individual bioactive components in these extracts, the results of the studies conducted are pointless and should be removed from the text together with Figures 6-8.

Answer: The purpose of the three stability tests is to evaluate the stability of phenolic compounds in methanol and ethanol extracts, because the total phenol content of the two extracts is the best among many solvent extracts. The changes in the content of phenolic compounds directly affect the antioxidant capacity of the extract, so TPheC and ABTS are used to evaluate the experimental results simultaneously. This is a relatively macroscopic method that can observe the desired results through changes in the overall content of phenolics. Although relatively rough, it can achieve the desired goal.

Question 12: 2.8. Could you explain how the methanolic extract of PK galls can be used as an antioxidant (stabilizing) additive for olive and sunflower oils intended for human consumption?

Answer: Thank you very much for your questions! We understand your meanings, both oils are consumed by humans, and the safety data of PK galls is too limited to be used as an antioxidant (stabilizing) additive. However, for this study, we only use olive oil or sunflower oil as the medium to study the stability of oil. It is not really necessary to commercialize the olive oil or sunflower oil added with PK galls, but only for research. The oxidative stabilities of oils are also one of the antioxidant assays in vitro, which is important for evaluating the antioxidant capacity of PK galls. At the same time, it is a potential application of PK galls. So I hope to reserve them.

Question 13: Table 9 is sufficient to show the results of cytotoxicity of methanolic and ethanolic extracts of PK insect galls. Figures 13 and 14 should be removed and included in the Supplementary Materials.

Answer: Your suggestion is very good! Figures 13 and 14 have been removed and included in the Supplementary Materials.

Question 14: 2.10. The authors wrote that: “a good correlation was found between the TCC values and its antioxidant activity (Figure 15). This implies that the carbohydrates in the methanolic extract are flavonoid glycosides such as luteolin-4'-O-glucoside” The problem is that this compound (luteolin-4'-O-glucoside) was not identified at all in the methanolic extract. And of course, it is not a carbohydrate!

Answer: Thank you very much for pointing out our mistake. Modifications have been made.

Question 15: My general comments are that a detailed qualitative and quantitative analysis for individual PK gall extracts should be done and the results presented. This is a prerequisite for further proper evaluation of the biological properties of the extracts and multivariate statistical analysis. Major correction and further peer-review evaluation of the obtained research results is essential.

Answer: Thank you very much for your valuable comments on our manuscript! We have also made the necessary modifications as much as possible. As mentioned earlier, the qualitative and quantitative analysis mentioned in this study is different from your requirements, we would complete this work in future research, but currently it involves doctoral graduation and there is not enough time to complete this work. We hope you could understand us.

Round 2

Reviewer 1 Report (New Reviewer)

All the modifications I asked was done 

Minor editing of English language required

Reviewer 2 Report (New Reviewer)

Since the Authors, for the most part, addressed the numerous comments of both reviewers, the manuscript can be accepted for publication in its current form.

The quality of English in the revised version is better, however, minor improvements would be valuable.

This manuscript is a resubmission of an earlier submission. The following is a list of the peer review reports and author responses from that submission.

Round 1

Reviewer 1 Report

The authors have carried out an extensive biological evaluation of insect gall extracts of Picea koraiensis, but the phytochemical profile was not performed rigorously enough. Although the advanced UPLC-MS technique applied in this work, the structural assignment is often not possible, e.g. why peak 4 is isoleucine and not leucine? Similarly, the structural details of the reported structures are not deductible from a simple analysis like the one made. It is not so evident why for some assignments [M + 2H] is reported as monocharged, eg. in octadecylamine m/z 270 [M + H]+ and also m/z 271 [M + 2H]+.

The references cited do not allow to understand the assignments, therefore the authors must comment whether each single component of the extract is present as an isolated metabolite according to the known literature. Moreover, MS and MS/MS analyses carried out in negative ion mode could give additional indications for a number of metabolites. Additionally, the tentatively assigned structures must be verified by comparison with a reference sample. A discussion on the polarity of the metabolites related to the retention time on reversed stationary phase must be also added. Peak 3 at line 142 cannot be protopine, which is an alkaloid with molecular composition C20H19NO5.

The structures reported in Figure 1 are too small and not all molecules indicated in chemical profile are shown ( i.e. not musolinic acid).

Furthermore, the authors must indicate which metabolites detected in the chemical profile can be responsible of the antioxidant activity, which are the triterpenoids, the phenols and flavonoids and metal chelators.

The Supporting file reports only qualitative tests which are not essential, except for didactic purposes. Instead, the authors must provide the results of the UPLC-MS analysis and the spectra by fragmentation experiments for each single component of the chemical profile.

Author Response

Question 1: The authors have carried out an extensive biological evaluation of insect gall extracts of Picea koraiensis, but the phytochemical profile was not performed rigorously enough. Although the advanced UPLC-MS technique applied in this work, the structural assignment is often not possible, e.g. why peak 4 is isoleucine and not leucine? Similarly, the structural details of the reported structures are not deductible from a simple analysis like the one made. It is not so evident why for some assignments [M + 2H] is reported as monocharged, eg. in octadecylamine m/z 270 [M + H]+ and also m/z 271 [M + 2H]+.

Answer: Dear reviewer, thank you very much for your recognition of our work. And thank you for your valuable comments to help us improve our manuscript. The extract was reanalyzed using Agilent 1290/6545 UHPLC-Q-TOF/MS, and corrected the existing errors according to your requirements.

Question 2:The references cited do not allow to understand the assignments, therefore the authors must comment whether each single component of the extract is present as an isolated metabolite according to the known literature. Moreover, MS and MS/MS analyses carried out in negative ion mode could give additional indications for a number of metabolites. Additionally, the tentatively assigned structures must be verified by comparison with a reference sample. A discussion on the polarity of the metabolites related to the retention time on reversed stationary phase must be also added. Peak 3 at line 142 cannot be protopine, which is an alkaloid with molecular composition C20H19NO5.

Answer: We have modified all the cited references to make them better evidence that our identification is correct. In addition, follow your suggestions, we also studied the extract in negative ion mode, but we found that the ionization of metabolites in negative ion mode was not ideal, so we still chose to identify the compounds in positive ion mode. Your suggestion is very good. However, due to limitations in experimental conditions and funding, there are still many metabolites that cannot be purchased as reference materials, therefore, we did not conduct comparative experiments. I hope you could understand us. A discussion on the polarity of the metabolites related to the retention time on reversed stationary phase has been added. Protopine is not included in the new metabolite analysis.

Question 3:The structures reported in Figure 1 are too small and not all molecules indicated in chemical profile are shown ( i.e. not musolinic acid).

Answer: The structure reported in Figure 1 has become larger and includes the structures of all molecules.

Question 4: Furthermore, the authors must indicate which metabolites detected in the chemical profile can be responsible of the antioxidant activity, which are the triterpenoids, the phenols and flavonoids and metal chelators.

Answer:Your suggestion is very good. After revising according to your requirements, the content is more complete. It could make readers have a better understanding of the relationship between the content of active ingredients and antioxidant activity. Thank you very much!

Question 5:The Supporting file reports only qualitative tests which are not essential, except for didactic purposes. Instead, the authors must provide the results of the UPLC-MS analysis and the spectra by fragmentation experiments for each single component of the chemical profile.

Answer: Thank you very much for your valuable suggestions. Pictures of qualitative tests have been deleted and the spectra by fragmentation experiments for each single component of the chemical profile have been added.

Reviewer 2 Report

The work is interesting but authors need to improve their manuscript so it can be of certain value.

1. Add picture of the authentically identified plant and the close-up of the galls, few of the galls.

2. The activity including the antioxidants etc. and yields needs be in double digits after the decimal.

3. The plants constituents reported by LC-MS analysis needs chromatographic comparisons with the standards, either co-injected or run at the same conditions. The chromatograms and the mass fragment patterns are not enough for definite identifications. Better would be to provide the fragments spectra of the compounds also.

4. The chemical constituents' tests are preliminary and not confirmative, many of them are cross-reactive; authors need to provide better and confirming tests!

5. The conclusions are speculative as for as the probable uses are concerned regarding the stability of the oils etc. 

Author Response

Question 1:The work is interesting but authors need to improve their manuscript so it can be of certain value. Add picture of the authentically identified plant and the close-up of the galls, few of the galls.

Answer: Dear reviewer, thank you very much for your recognition of our work. And thank you for your valuable comments to help us improve our manuscript. Your suggestions are very good. Relevant pictures have been added.

Question 2:The activity including the antioxidants etc. and yields needs be in double digits after the decimal.

Answer: We have revised them as you request.

Question 3: The plants constituents reported by LC-MS analysis needs chromatographic comparisons with the standards, either co-injected or run at the same conditions. The chromatograms and the mass fragment patterns are not enough for definite identifications. Better would be to provide the fragments spectra of the compounds also.

Answer: Your suggestion is very good. However, due to limitations in experimental conditions and funding, there are still many metabolites that cannot be purchased as reference materials, therefore, we did not conduct comparative experiments. I hope you could understand us. The fragments spectra of the compounds have been provided according to your requirements.

Question 4:The chemical constituents' tests are preliminary and not confirmative, many of them are cross-reactive; authors need to provide better and confirming tests!

Answer: Preliminary tests of chemical constituents are foundational tests and it really exists that you say. We therefore designed at least two types of experiments for each class of chemical constituents at the time we designed our experiments with the goal of avoiding false-positive and false negative results. At the same time, the constituents whose results were positive were tested for content determination using spectrophotometry, and all had good results.

Question 5:The conclusions are speculative as for as the probable uses are concerned regarding the stability of the oils etc.

Answer: We completed the study on the oil stability of methanol extract, the aim of these experiments was to explore possible applications of the methanolic extract. In the conclusion section, we also say that it has the potential to act as an oily stabilizer.

Round 2

Reviewer 1 Report

In their revised version, the authors have significantly improved the results, but still some improvements and clarifications can be introduced.In particular:

a) regarding data reported in table 2, on page 5 only the results from extractions was commented, while the remaining data (TCC, TPC, TTC) are discussed much later in the text: it is better to report them at the time of the discussion, i.e. including them in table 4.

 b) in par. 2.3 the authors should briefly discuss the choice of investigating the methanol extract and not the others.

 c) In all the tables the meaning of the letters a, b etc. is not clear; they must be specified as footnotes

d) in table 3, the identification of [M+NH4]+ for peaks 12 and 13 should be commented

e) for the components of peaks 13 and 19, the stereochemistry 16 alpha e (10e,12e, 15e) is not supported by comparison with standards; therefore, it is better to indicate "tentatively identified"

f) table 4 must be reported after citing it in the text.

g) at page 4, high isoorientin content: based on what could the content be established high? why not also gossypitrin which contains the same ortho-hydroxyphenol moiety?

h) report the correct name of gossypitrin in figure 2

 j) UPLC-MS analysis is only performed for methanol extract, therefore you could not be sure that the ethanol extract contains these metabolites; it must be indicated hypothetically,  i.e. it is assumed that ..

k) remove TBHQ from table 6and L-ascorbic acid from Table 7, because data of these standards are not reported

l) In Table 8 and in the text, replace with the correct abbreviations AChE and BChE

Author Response

Question 1: In their revised version, the authors have significantly improved the results, but still some improvements and clarifications can be introduced.In particular:

Answer: Dear reviewer, thank you very much for your recognition of our work. And thank you for your valuable comments to help us improve our manuscript. We revised all questions according to your requirements.

Question 2: a) regarding data reported in table 2, on page 5 only the results from extractions was commented, while the remaining data (TCC, TPC, TTC) are discussed much later in the text: it is better to report them at the time of the discussion, i.e. including them in table 4.

Answer: Thank you very much for your valuable suggestions. It has been revised according to your suggestions.

Question 3: b) in par. 2.3 the authors should briefly discuss the choice of investigating the methanol extract and not the others.

Answer: Your suggestion is very good! After revising according to your requirements, the content is more complete, it may stimulate readers' interest in reading. But we think this paragraph is placed in para 2.2 is more appropriate. So we put it in para 2.2.

Question 4: c) In all the tables the meaning of the letters a, b etc. is not clear; they must be specified as footnotes

Answer: They have been added in the text as required.

Question 5: d) in table 3, the identification of [M+NH4]+ for peaks 12 and 13 should be commented

Answer: Comments have been completed as required.

Question 6: e) for the components of peaks 13 and 19, the stereochemistry 16 alpha e (10e,12e, 15e) is not supported by comparison with standards; therefore, it is better to indicate "tentatively identified"

Answer: Thank you very much! According to your suggestions, the revised content is more accurate and rigorous.

Question 7: f) table 4 must be reported after citing it in the text.

Answer: It has been revised according to your requirement.

Question 8: g) at page 4, high isoorientin content: based on what could the content be established high? why not also gossypitrin which contains the same ortho-hydroxyphenol moiety?

Answer: Your question is very professional. Our previous description was inaccurate and has been revised. Thank you very much for your reminder! Gossypitrin is also a good antioxidant. We added it in the text and cited the literature.

Question 9: h) report the correct name of gossypitrin in figure 2

Answer: Dear reviewer, we carefully checked its name in Figure 2, and no error was found.

Question 10: j) UPLC-MS analysis is only performed for methanol extract, therefore you could not be sure that the ethanol extract contains these metabolites; it must be indicated hypothetically,  i.e. it is assumed that ..

Answer: Thank you very much for your valuable suggestions. It has been revised according to your suggestions.

Question 11: k) remove TBHQ from table 6and L-ascorbic acid from Table 7, because data of these standards are not reported

Answer: They have been removed according to your suggestions.

Question 12: l) In Table 8 and in the text, replace with the correct abbreviations AChE and BChE

Answer: All errors in the text have been corrected.